# One-Shot Heterogeneous Federated Learning with Local Model-Guided Diffusion Models

**Mingzhao Yang**[1]  **Shangchao Su**[1]  **Bin Li**[*1]  **Xiangyang Xue**[1]

## Abstract

In recent years, One-shot Federated Learning (OSFL) methods based on Diffusion Models (DMs) have garnered increasing attention due to their remarkable performance. However, most of these methods require the deployment of foundation models on client devices, which significantly raises the computational requirements and reduces their adaptability to heterogeneous client models. In this paper, we propose FedLMG, a heterogeneous one-shot Federated learning method with Local Model-Guided diffusion models. In our method, clients do not need access to any foundation models but only train and upload their local models, which is consistent with traditional FL methods. On the clients, we employ classification loss and batch normalization loss to capture the broad category features and detailed contextual features of the client distributions. On the server, based on the uploaded client models, we utilize backpropagation to guide the server's DM in generating synthetic datasets that comply with the client distributions, which are then used to train the aggregated model. By using the local models as a medium to transfer client knowledge, our method significantly reduces the computational requirements on client devices and effectively adapts to scenarios with heterogeneous clients. Extensive quantitation and visualization experiments on three large-scale real-world datasets, along with theoretical analysis, demonstrate that the synthetic datasets generated by FedLMG exhibit comparable quality and diversity to the client datasets, which leads to an aggregated model that outperforms all compared methods and even the performance ceiling, further elucidating the significant potential of utilizing DMs in FL.

[1]Shanghai Key Laboratory of Intelligent Information Processing, School of Computer Science, Fudan University. Correspondence to: Bin Li <libin@fudan.edu.cn>.

*Proceedings of the 42nd International Conference on Machine Learning*, Vancouver, Canada. PMLR 267, 2025. Copyright 2025 by the author(s).

## 1. Introduction

Federated learning (FL) (Mammen, 2021) has gained increasing attention recently. Standard FL relies on frequent communication between the server and clients. With the growing adoption of AI models by individual users, the application scenarios of FL have expanded significantly, including mobile photo album categorization and autonomous driving (Nguyen et al., 2022; Fantauzzo et al., 2022). However, in these scenarios, the substantial communication cost associated with FL is often impractical for individual users. As a result, one-shot FL (OSFL) has emerged as a solution (Li et al., 2021; Zhou et al., 2020). OSFL aims to establish the aggregated model within a single communication round. Currently, mainstream OSFL methods can be categorized into four types: 1) Methods using the auxiliary dataset (Guha et al., 2019; Li et al., 2020a; Lin et al., 2020). 2) Methods training generators (Zhang et al., 2022; Heinbaugh et al., 2022). 3) Methods transferring auxiliary information (Zhou et al., 2020; Su et al., 2023). 4) Methods based on DMs (Yang et al., 2024a;b; Zhang et al., 2023a).

However, existing methods are hard to apply in real-world scenarios due to the following reasons: 1) Collecting public datasets that comply with all client distribution is impractical, owing to privacy concerns and data diversity issues. 2) Due to the limited computational power and data of the clients, training generators on realistic client images is challenging. So most OSFL methods can only be applied to small-scale toy datasets, such as MNIST and CIFAR10. 3) The transmission of auxiliary information incurs communication cost, and extracting the auxiliary information on the clients also incurs additional computation cost, further restricting the practicality of OSFL. 4) Current OSFL methods that leverage DMs necessitate the deployment of foundation models on the clients (Yang et al., 2024a; Zhang et al., 2023a), such as CLIP (Radford et al., 2021) and BLIP (Li et al., 2023), or directly involve client training of diffusion models (Yang et al., 2024b), leading to significant communication and computational costs. In comparison to widely used traditional FL methods, which only require local training of client models, these additional burdens imposed by OSFL methods significantly increase the strain on client resources, thereby limiting their practical applicability.

To reduce the computational requirements on client devices and alleviate the burden on clients, we propose FedLMG, a heterogeneous one-shot **Fed**erated learning method with **L**ocal **M**odel-**G**uided diffusion models. Our method involves using the locally trained client models to guide the DMs in generating the synthetic dataset that complies with different client distributions. Specifically, FedLMG consists of three steps: Local Training, Image Generation, and Model Aggregation. Firstly, based on the theoretical analyses of the client's local distribution and the server's conditional distribution, the clients independently train the client models on their private data and upload them to the server. Subsequently, assisted by the received client models, the server generates realistic images that comply with different client distributions based on DMs. After obtaining the synthetic dataset, we introduce three strategies to obtain the aggregated model: fine-tuning, multi-teacher distillation, and specific-teacher distillation. Through these strategies, we achieve the model aggregation in a single round of communication, without accessing any client data or any additional information transferring compared with traditional FL.

To validate the performance of our method, we conduct extensive quantitation and visualization experiments on three large-scale real image datasets: DomainNet (Peng et al., 2019), OpenImage (Kuznetsova et al., 2020) and NICO++ (Zhang et al., 2023c). Sufficient quantitation experiments under various client scenarios demonstrate that our method outperforms all compared methods in a single communication round, and in some cases even surpasses the ceiling performance of centralized training, strongly underscoring the potential of DMs and providing convincing evidence for our aforementioned ideas. Visualization experiments also illustrate that our method generates synthetic datasets that comply with both the specific categories and the personalized client distributions, with comparable quality and diversity to the original client dataset. Moreover, we conduct thorough discussions on communication cost, computational cost, and privacy concerns, further enhancing the practicality of the proposed method.

In summary, this paper makes the following contributions:

- We propose FedLMG, a novel OSFL method, to achieve real-world OSFL without utilizing any foundation models on the clients, ensuring no additional communicational or computational burden compared to traditional FL methods, thereby significantly expanding the practicality of OSFL.

- We propose using the locally trained client models as a medium to transfer client knowledge to the server, guiding the diffusion model through classification loss and BN loss to capture the client's category and contextual features, thereby generating high-quality synthetic datasets that comply with client distributions.

- We conduct thorough theoretical analyses, demonstrating that under the assistance of client models, the KL divergence between the data distribution of the DM on the server and the local data distribution is bounded.

- We conduct sufficient quantitation and visualization experiments to demonstrate that the proposed method outperforms other compared methods and can even surpass the performance ceiling of centralized training in some cases, further evidencing the enormous potential of utilizing DM in OSFL.

## 2. Related Work

### 2.1. One-shot Federated Learning

In the standard FL (McMahan et al., 2017), there are multiple rounds of communication between the server and clients. To reduce the high communication cost, OSFL entails clients training their local models to convergence first, followed by aggregation on the server. Existing OSFL methods can be broadly categorized into three main types. 1) Methods based on public auxiliary dataset. (Guha et al., 2019) utilizes unlabeled public data on the server for model distillation. Similarly, FedKT (Li et al., 2020a) and FedDF (Lin et al., 2020) employ an auxiliary dataset for knowledge transfer on the server. 2) Methods based on generators. DENSE (Zhang et al., 2022) employs an ensemble of client models as a discriminator to train a generator for generating pseudo samples, which is used to train the aggregated model. To address very high statistical heterogeneity, FedCVAE (Heinbaugh et al., 2022) trains a conditional variational autoencoder (CVAE) on the client side and sends the decoders to the server to generate data. 3) Methods based on sharing auxiliary information. DOSFL (Zhou et al., 2020) performs data distillation on the client, and the distilled pseudo samples are uploaded to the server for global model training. MAEcho (Su et al., 2023) shares the orthogonal projection matrices of client features to the server to optimize global model parameters. 4) Methods based on DMs. We provide a detailed introduction to such methods in the next section. Although a large number of OSFL methods have been proposed, their practicality is significantly limited due to the reasons mentioned in the Introduction.

### 2.2. FL with Diffusion Models

The DM is introduced by (Sohl-Dickstein et al., 2015). (Ho et al., 2020) proposes the fundamental framework of the DM. Following this, a series of sampling techniques emerge (Song et al., 2020a; Liu et al., 2022; Song et al., 2020b), leading to the success of DMs in generation (Kingma et al., 2021; Wang et al., 2018). Subsequently, Stable Diffusion (Rombach et al., 2022) provides a series of powerful DM capable of generating images complying with the most common

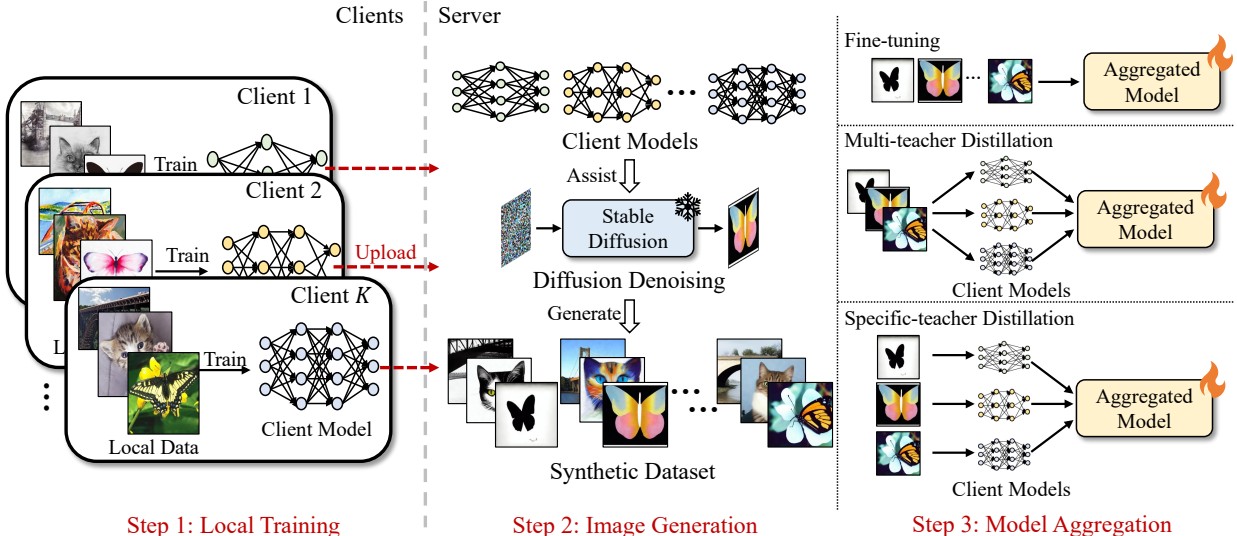

*Figure 1.* Overview of FedLMG. Our method consists of three steps: **Local Training**, **Image Generation**, and **Model Aggregation**. Firstly, each client independently trains their models using its private data and uploads them to the server. Assisted by these client models, our method leverages the powerful DM to obtain the synthetic dataset that complies with different client distributions. Based on the synthetic dataset, three strategies are provided to obtain the aggregated model.

real-world distributions. One major advantage of DMs lies in their powerful conditional generation capability. By using images (Saharia et al., 2022a; Wang et al., 2022; Zhang et al., 2023b), text (Nichol et al., 2022; Saharia et al., 2022b; Kim et al., 2022; Preechakul et al., 2022), other models (Dhariwal & Nichol, 2021), or initial noise (Mao et al., 2023) as condition, DMs can accordingly generate high-quality images, inspiring the development of DM-based FL methods.

Currently, only a small number of studies have focused on the significant potential of DMs in FL. Most DM-based FL methods (Yang et al., 2024a;b) are based on the idea of uploading distribution information of the client data to the server, which can be traced back to (Li et al., 2007). In FedDISC (Yang et al., 2024a), Stable Diffusion is introduced into semi-supervised FL for the first time, achieving remarkable results within just a single communication round. However, it requires using the CLIP image branch as the backbone for classification, which limits its flexibility. FGL (Zhang et al., 2023a) employ BLIPv2 (Li et al., 2023) on the clients to extract text prompts of client images, which are then sent to the server for image generation using DMs. FedDEO (Yang et al., 2024b) trains a description of the local distribution using DMs on the clients. Phoenix (Stanley Jothiraj & Mashhadi, 2024) introduced FL into DMs, proposing a distributed method for training DMs. All the aforementioned methods involve substantial computation and communication costs on the clients. Unlike these methods, we do not require deploying any foundational models on the clients, which significantly reduces communication and computational costs on the clients, further enhancing

the practicality of our method and enabling our method to address the scenario of heterogeneous client models.

## 3. Method

In this section, we introduce the proposed method in detail. Firstly, we provide essential preliminaries regarding the diffusion process and our problem setting. Then, as stated in Figure 1, we detail the three steps of the proposed method: Local Training, Image Generation, and Model Aggregation.

### 3.1. Preliminaries

**Problem Setting and Notations.** Consider that we have $K$ clients. Taking client $k$ as an example, this client has a private dataset $\mathcal{D}^k = \{(\mathbf{x}_i, y_i)\}_{i=1}^{N_k}$ and a client model $\mathcal{F}_{\boldsymbol{\theta}_k}$. The server needs to aggregate the client models $\{\mathcal{F}_{\boldsymbol{\theta}_k}\}_{k=1}^{K}$ to obtain an aggregated model $\mathcal{F}_{\boldsymbol{\theta}_g}$ that adapts various client distributions. The overall objective of our method is:

$$\min_{\mathbf{w} \in \mathbb{R}^d} \frac{1}{K} \sum_{k=1}^{K} \mathbb{E}_{\mathbf{x} \sim \mathcal{D}_k} \left[ \ell_k(\mathcal{F}_{\boldsymbol{\theta}_g}; \mathbf{x}) \right] \tag{1}$$

where $\ell_k$ is the local objective function for the $k$-th client, $\mathcal{F}_{\boldsymbol{\theta}_g}$ is the parameters of the aggregated model. From this objective function, it is evident that our goal is to train an aggregated model that adapts to all client distributions and exhibits excellent classification performance on the data from each client. We evaluate the performance in the subsequent experimental section according to this objective.

**Diffusion Process.** The DM $\epsilon_\theta$ samples initial noise $\mathbf{x}_T$

from a standard Gaussian Distribution $\mathcal{N}(0, \mathcal{I})$ and iteratively denoises it, resulting in a realistic image $\mathbf{x}_0$, where $T$ denotes the maximum timestep. (Dhariwal & Nichol, 2021) proposes the classifier-guidance, wherein the gradients backpropagated through the classifier $\mathcal{F}_{\boldsymbol{\theta}_k}$ are used to modify the predicted noise of the DM $\epsilon_\theta(\mathbf{x}_t, t)$ at each timestep. The loss function utilized for generating gradients typically involves the cross-entropy loss $\mathcal{L}_{CE}$ between the given class label $y$ and the output of classifier $\mathcal{F}_{\boldsymbol{\theta}_k}(\mathbf{x}_t)$. Under the guidance of the classifier $\mathcal{F}_{\boldsymbol{\theta}_k}$, the sampling process at each timestep has two steps: Firstly, for any given timestep $t \in \{0, \ldots, T\}$, the predicted noise is modified by the gradients of $\mathcal{F}_{\boldsymbol{\theta}_k}$ according to the following equation:

$$\hat{\epsilon}(\mathbf{x}_t, t|y) := \epsilon_\theta(\mathbf{x}_t, t|y) - \sqrt{1 - \bar{\alpha}_t} \nabla_{\mathbf{x}_t} \log p_{\boldsymbol{\theta}_k}(y|\mathbf{x}_t) \quad (2)$$

where $\nabla_{\mathbf{x}_t} \log p_{\boldsymbol{\theta}_k}(y|\mathbf{x}_t)$ is the gradient of the classifier $\mathcal{F}_{\boldsymbol{\theta}_k}$ with respect to the input $(\mathbf{x}_t, y)$. Afterwards, utilizing the modified $\hat{\epsilon}(\mathbf{x}_t, t|y)$, the sample for the next time step $\mathbf{x}_{t-1}$ is obtained:

$$\begin{aligned}\mathbf{x}_{t-1} =& \sqrt{\alpha_{t-1}} \left( \frac{\mathbf{x}_t - \sqrt{1 - \alpha_t}\hat{\epsilon}(\mathbf{x}_t, t|y)}{\sqrt{\alpha_t}} \right) \\ &+ \sqrt{1 - \alpha_{t-1} - \sigma_t^2} \cdot \hat{\epsilon}(\mathbf{x}_t, t|y) + \sigma_t \varepsilon_t \quad (3)\end{aligned}$$

where $\alpha_t$, $\alpha_{t-1}$ and $\sigma_t$ are pre-defined parameters, $\varepsilon_t$ is the Gaussian noise randomly sampled at each timestep. According to the diffusion process described above, for the randomly sampled initial noise $\mathbf{x}_T \sim \mathcal{N}(0, \mathcal{I})$, after $T$ iterations, the diffusion model can denoise the initial noise into high-quality realistic images $\mathbf{x}_0$ under the guidance of the classifier $\mathcal{F}_{\boldsymbol{\theta}_k}$.

## 3.2. Local Training

The first step of our method is local training. To generate data on the server that complies with the client distribution, we need to transmit client information to the server. In federated learning, it's common to send locally trained client models to the server. Therefore, for the local data distribution $p_k(\mathbf{x})$ of the private dataset $\mathcal{D}^k$ for client $k$, we leverage the information from the locally trained client model $\mathcal{F}_{\boldsymbol{\theta}_k}$ on the server and obtain the conditional distribution $p_{\epsilon_\theta}(\mathbf{x}|\boldsymbol{\theta}_k)$ based on the data distribution of the diffusion model $p_{\epsilon_\theta}(\mathbf{x})$. We aim for the synthetic dataset sampled from the conditional distribution $p_{\epsilon_\theta}(\mathbf{x}|\boldsymbol{\theta}_k)$ to closely comply with the client local data distribution $p_k(\mathbf{x})$. Therefore, considering the relationship between $p_k(\mathbf{x})$ and $p_{\epsilon_\theta}(\mathbf{x}|\boldsymbol{\theta}_k)$ is essential. We conduct comprehensive theoretical analyses regarding the relationship between these two distributions.

Firstly, we need to analyze the relationship between the unconditional data distribution of diffusion model $p_{\epsilon_\theta}(\mathbf{x})$ and the client local data distribution $p_k(\mathbf{x})$. As stated in the Introduction, our motivation for utilizing the diffusion

model lies in its ability to generate data that comply with almost any data distribution with proper guidance. Therefore, regarding the data distribution $p_k(\mathbf{x})$ of the client's local dataset $\mathcal{D}^k$ and the data distribution $p_{\epsilon_\theta}(\mathbf{x})$ that the DMs $\epsilon_\theta$ can generate, we can make the following assumption:

**Assumption 1** *There exists $\lambda > 0$ such that the Kullback-Leibler divergence from $p_k(\mathbf{x})$ to $p_{\epsilon_\theta}(\mathbf{x})$ is bounded above by $\lambda$:*

$$KL(p_{\epsilon_\theta}(\mathbf{x})\|p_k(\mathbf{x})) < \lambda \quad (4)$$

In this assumption, we assume that there is some overlap between $p_k(\mathbf{x})$ and $p_{\epsilon_\theta}(\mathbf{x})$, which is considered reasonable. We don't rigidly demand that $p_{\epsilon_\theta}(\mathbf{x})$ fully encompasses $p_k(\mathbf{x})$. Even if the clients specialize in certain professional domains such as medical images, given the widespread application of DMs across various domains (Kazerouni et al., 2023; Wang et al., 2024), the assumption can be satisfied by replacing the DM being used. Based on Assumption 1, we have the following theorem regarding the relationship between the conditional distribution $p_{\epsilon_\theta}(\mathbf{x}|\boldsymbol{\theta}_k)$ and the client local data distribution $p_k(\mathbf{x})$:

**Theorem 1** *There exists $\lambda > 0$, for the local data distribution $p_k(\mathbf{x})$ and the conditional distribution $p_{\epsilon_\theta}(\mathbf{x}|\boldsymbol{\theta}_k)$ of the DM $\epsilon_\theta$ conditioned on the client model $\mathcal{F}_{\boldsymbol{\theta}_k}$ trained on client $k$, we have:*

$$\begin{aligned}KL(p_k(\mathbf{x})\|p_{\epsilon_\theta}(\mathbf{x}|\boldsymbol{\theta}_k)) <& \lambda + \mathbb{E}(\log p_{\epsilon_\theta}(\boldsymbol{\theta}_k)) \\ &- \int p_k(\mathbf{x}) \log p_{\epsilon_\theta}(\boldsymbol{\theta}_k|\mathbf{x}) d\mathbf{x} \quad (5)\end{aligned}$$

For a detailed proof, please refer to the appendix. From Eq. 5, we can observe that the KL divergence between the conditional distribution $p_{\epsilon_\theta}(\mathbf{x}|\boldsymbol{\theta}_k)$, which is also the distribution of the synthetic data, and the distribution of client's local data $p_k(\mathbf{x})$ is bounded above. This upper bound consists of three components: $\lambda$, $\mathbb{E}(\log p_{\epsilon_\theta}(\boldsymbol{\theta}_k))$, and $- \int p_k(\mathbf{x}) \log p_{\epsilon_\theta}(\boldsymbol{\theta}_k|\mathbf{x})d\mathbf{x}$. $\lambda$ is the same as in Eq. 4. $\mathbb{E}(\log p_{\epsilon_\theta}(\boldsymbol{\theta}_k))$ is a constant independent of the sample $\mathbf{x}$. $- \int p_k(\mathbf{x}) \log p_{\epsilon_\theta}(\boldsymbol{\theta}_k|\mathbf{x})d\mathbf{x}$ is the negative log-likelihood between the client model $\mathcal{F}_{\boldsymbol{\theta}_k}$ and the client distribution $p_k(\mathbf{x})$. Minimizing this negative log-likelihood is equivalent to minimizing the cross-entropy loss $\mathcal{L}_{CE}$. This implies that during the local training, we need to train client models using the cross-entropy loss to minimize this upper bound of the KL divergence. Consequently, the conditional distribution of the synthetic dataset $p_{\epsilon_\theta}(\mathbf{x}|\boldsymbol{\theta}_k)$ can closely approximate the client's local data distribution $p_k(\mathbf{x})$. Therefore, for client $k$, we utilize its privacy dataset $\mathcal{D}^k$ and train the client model $\mathcal{F}_{\boldsymbol{\theta}_k}$ using the following cross-entropy loss function:

$$\ell_k(\mathcal{F}_{\boldsymbol{\theta}_g}; \mathbf{x}) = \mathcal{L}_{CE}(\mathcal{F}_{\boldsymbol{\theta}_k}(\mathbf{x}_i^k), y_i^k) \quad (6)$$

After multiple rounds of training, the client models are sent to the server to guide the generation process on the server. It's worth noting that in FedLMG, there are no requirements for the used model structures, which further enhances the practicality of our method.

### 3.3. Image Generation

After receiving the locally trained client models $\mathcal{F}_{\boldsymbol{\theta}_k}, k = 1, \ldots, K$ uploaded by the clients, these client models serve as cues for the DM, generating the synthetic dataset complies with different client distributions. Firstly, we elaborate on how the client models assist DM in generation. In our problem setting, generated images must possess accurate categories and comply with specified client distributions, introducing novel demands to sampling, and necessitating consideration of additional image attributes such as style, color, background, etc. Relying solely on classification results falls significantly short of achieving these demands since the classification results mainly provide information on categories.

To provide detailed context information about the client distributions, we utilize the statistics of each batch normalization (BN) layer of client models $\mathcal{F}_{\boldsymbol{\theta}_k}$: mean $\mu$ and variance $\sigma$. In other words, we need to consider the conditional generation process $p(\mathbf{x}_{t-1}|\mathbf{x}_t, y, \{\boldsymbol{\mu}_{k,l}\}_{l=1}^{L_k}, \{\boldsymbol{\sigma}_{k,l}\}_{l=1}^{L_k})$, where $\boldsymbol{\mu}_{k,l}$ and $\boldsymbol{\sigma}_{k,l}$ respectively denote the means and the variances of all BN layers within $\mathcal{F}_{\boldsymbol{\theta}_k}$, and $L_k$ represents the number of BN layers within $\mathcal{F}_{\boldsymbol{\theta}_k}$. Therefore, during modifying the predicted noise of the DM $\epsilon_\theta(\mathbf{x}_t, t|y)$ at each time step $t$, we compute gradients by summing the cross-entropy loss $\mathcal{L}_{CE}$ and the BN Loss $\mathcal{L}_{BN}$ to incorporate the additional distribution details embedded within the statistics of the BN layers into the diffusion process. The computation of $\mathcal{L}_{BN}$ is as follows:

$$\mathcal{L}_{BN} = \sum_{l=1}^{L}(\|\boldsymbol{\mu}_l(\mathbf{x}, \boldsymbol{\theta}_k) - \boldsymbol{\mu}_{k,l}\| + \|\boldsymbol{\sigma}_l^2(\mathbf{x}, \boldsymbol{\theta}_k) - \boldsymbol{\sigma}_{k,l}^2\|)$$
$$(7)$$

where $\boldsymbol{\mu}_l(\mathbf{x}, \boldsymbol{\theta}_k)$ and $\boldsymbol{\sigma}_l^2(\mathbf{x}, \boldsymbol{\theta}_k)$ denote the mean and variance of the output feature from the $l$-th BN layer after feeding the sample $\mathbf{x}$ into the client model $\mathcal{F}_{\boldsymbol{\theta}_k}$.

Furthermore, since local models $\mathcal{F}_{\boldsymbol{\theta}_k}$ are simply trained on the client and are not accustomed to the noised input $\mathbf{x}_t$, traditional classifier-guidance struggles to provide accurate guidance through the computed gradient $\nabla_{\mathbf{x}_t} \log p_{\boldsymbol{\theta}_k}(y|\mathbf{x}_t)$. To address this challenge, at any time step $t$, we utilize the predicted noise $\epsilon_\theta(\mathbf{x}_t, t|y)$ to predict $\mathbf{x}_0$ according to the following equation:

$$\hat{\mathbf{x}}_{0,t} = \frac{\mathbf{x}_t - \sqrt{1 - \alpha_t}\hat{\epsilon}(\mathbf{x}_t, t|y)}{\sqrt{\alpha_t}}$$
$$(8)$$

Subsequently, based on $\hat{\mathbf{x}}_{0,t}$, we compute the loss function and gradient to modify $\epsilon_\theta(\mathbf{x}_t, t|y)$. Although in the initial time steps, $\hat{\mathbf{x}}_{0,t}$ may appear blurry, the noise level in comparison to $\mathbf{x}_t$ is noticeably reduced. This decreases the demand for the client models' robustness of noise, mitigating the need for clients to specifically train for classifying noised samples and further enhancing the practicality of our method.

Finally, the overall loss function $\mathcal{L}$ employed in the conditional generation is as follows:

$$\mathcal{L}(\mathbf{x}_t, y, \boldsymbol{\theta}_k) = \mathcal{L}_{CE}(\mathcal{F}_{\boldsymbol{\theta}_k}(\hat{\mathbf{x}}_{0,t}), y) + w_{BN}\mathcal{L}_{BN}(\hat{\mathbf{x}}_{0,t}, \boldsymbol{\theta}_k)$$
$$(9)$$

where $w_{BN}$ is the weight of BN Loss. After performing gradient backpropagation according to this loss function, we use Eqs. 2 and 3 to guide the generation process through the client model $\mathcal{F}_{\boldsymbol{\theta}_k}$ and its accompanying BN statistics, enabling conditional generation.

For any given time step $t \in \{0, \ldots, T\}$, we modify the predicted noise of the DM based on Eq. 9:

$$\hat{\epsilon}(\mathbf{x}_t, t|y) := \epsilon_\theta(\mathbf{x}_t, t|y) - \sqrt{1 - \bar{\alpha}_t}\nabla_{\mathbf{x}_t}\mathcal{L}(\mathbf{x}_t, y, \boldsymbol{\theta}_k) \quad (10)$$

Subsequently, based on Eq. 3, we compute $\mathbf{x}_{t-1}$ using the modified $\hat{\epsilon}(\mathbf{x}_t, t|y)$, leading to the realistic image $\mathbf{x}_0$ after $T$ iterations. During the generation, since we specify the category $y$ and the classifier $\mathcal{F}_{\boldsymbol{\theta}_k}$, the generated image $\mathbf{x}_0$ is automatically labeled. We define $\mathbf{x}_0$ as $\hat{\mathbf{x}}_i^k$ and include along with its label $y_i^k$ in the synthetic dataset $\hat{\mathbf{X}}$. After undergoing multiple iterations of generation, we obtain the synthetic dataset $\hat{\mathbf{X}} = \{(\hat{\mathbf{x}}_i^k, y_i^k)\}_{i=1}^N$.

### 3.4. Model Aggregation

Based on the synthetic dataset $\hat{\mathbf{X}}$, we proceed to obtain the aggregated model. We employ the idea of distillation to achieve model aggregation and introduce three strategies to obtain the aggregated model: Fine-tuning, Multi-teacher Distillation, and Specific-teacher Distillation.

**Fine-tuning.** As all samples $\hat{\mathbf{x}}_i^k, i = 1, \ldots, N, k = 1, \ldots, K$ in the synthetic dataset $\hat{\mathbf{X}}$ have their labels $y_i^k$, this strategy refers to directly training an aggregated model using the cross-entropy loss $\mathcal{L}_{CE}$. The specific loss function used during the aggregation process is as follows:

$$\mathcal{L}_{agg}(\hat{\mathbf{x}}_i^k, y_i^k) = \mathcal{L}_{CE}(\mathcal{F}_{\boldsymbol{\theta}_g}(\hat{\mathbf{x}}_i^k), y_i^k) \quad (11)$$

Although this strategy is relatively simple, since the performance ceiling of centralized training also involves training the aggregated model on the client data, we want to emphasize that this strategy is closest to the centralized training, directly reflecting the quality and diversity difference between the synthetic dataset and the original client dataset.

**Multi-teacher Distillation.** The second strategy is multi-teacher distillation. The synthetic dataset $\hat{\mathbf{X}}$ serves as a

*Table 1.* Performance of different methods on OpenImage, DomainNet, Unique NICO++, and Common NICO++ under feature distribution skew, where italicized text represents the ceiling performance used solely as a reference, and bold text signifies the optimal performance excluding the ceiling performance.

| | OpenImage | | | | | | | DomainNet | | | | | | |
| | client0 | client1 | client2 | client3 | client4 | client5 | Avg | clipart | infograph | painting | quickdraw | real | sketch | Avg |
|---|---|---|---|---|---|---|---|---|---|---|---|---|---|---|
| *Ceiling* | *49.88* | *50.56* | *57.89* | *59.96* | *66.53* | *51.38* | *56.03* | *47.48* | *19.64* | *45.24* | *12.31* | *59.79* | *42.35* | *36.89* |
| FedAvg | 41.46 | 50.36 | 52.61 | 50.36 | 62.10 | 50.17 | 51.18 | 37.96 | 12.55 | 34.41 | 5.93 | 51.33 | 32.37 | 29.09 |
| FedDF | 44.96 | 46.15 | 59.69 | 58.69 | 63.45 | 46.63 | 53.26 | 38.09 | 13.68 | 35.48 | 7.32 | 53.83 | 34.69 | 30.52 |
| FedProx | 44.99 | 48.83 | 49.25 | 56.68 | 61.23 | 46.07 | 51.18 | 38.24 | 12.46 | 37.29 | 6.26 | 54.88 | 35.76 | 30.82 |
| FedDyn | 46.93 | 46.08 | 52.44 | 54.67 | 62.84 | 47.73 | 51.78 | 40.12 | 14.77 | 36.59 | 7.73 | 54.85 | 34.81 | 31.48 |
| Prompts Only | 30.41 | 30.23 | 42.92 | 43.48 | 50.75 | 33.43 | 38.54 | 31.8 | 11.61 | 31.14 | 4.13 | **61.53** | 31.44 | 28.61 |
| FedDISC | 47.42 | 49.65 | 54.73 | 53.41 | 60.74 | 52.81 | 53.13 | 43.89 | 14.84 | 38.38 | 8.35 | 56.19 | 36.82 | 33.08 |
| FGL | 48.21 | 49.16 | 54.98 | 55.47 | 63.14 | 49.32 | 53.38 | 41.81 | 15.30 | 40.67 | 8.79 | 57.58 | 39.54 | 33.95 |
| FedLMG_FT | **48.99** | 51.66 | 55.59 | 52.80 | 62.41 | 58.86 | 55.05 | 44.25 | 17.51 | 38.74 | 9.43 | 57.31 | 38.44 | 34.28 |
| FedLMG_SD | 47.60 | **55.20** | **61.54** | **61.83** | **67.07** | **59.90** | **58.86** | **46.23** | 18.42 | **42.85** | **10.24** | 58.52 | 39.13 | 35.90 |
| FedLMG_MD | 44.70 | 53.08 | 58.67 | 60.13 | 64.06 | 58.06 | 56.45 | 47.21 | **18.49** | 40.37 | 10.02 | 59.67 | **40.19** | **35.99** |
| | Unique NICO++ | | | | | | | Common NICO++ | | | | | | |
| | client0 | client1 | client2 | client3 | client4 | client5 | Avg | autumn | dim | grass | outdoor | rock | water | Avg |
| *Ceiling* | *79.16* | *81.51* | *76.04* | *72.91* | *79.16* | *79.29* | *78.01* | *62.66* | *54.07* | *64.89* | *63.04* | *61.08* | *54.63* | *60.06* |
| FedAvg | 67.31 | 74.73 | 69.01 | 64.37 | 73.07 | 67.87 | 69.39 | 52.51 | 40.45 | 57.21 | 51.59 | 49.31 | 43.56 | 49.11 |
| FedDF | 69.79 | 78.90 | 69.53 | 66.01 | 74.86 | 70.80 | 71.65 | 50.44 | 39.62 | 57.42 | 52.91 | 51.61 | 44.76 | 49.46 |
| FedProx | 70.46 | 75.3 | 70.87 | 67.67 | 72.84 | 71.51 | 71.44 | 53.49 | 42.41 | 58.84 | 53.08 | 53.67 | 45.42 | 51.15 |
| FedDyn | 71.23 | 74.98 | 69.68 | 68.13 | 73.63 | 70.61 | 71.37 | 54.38 | 43.20 | 57.56 | 52.63 | 52.86 | 46.76 | 51.23 |
| Prompts Only | 69.79 | 69.14 | 69.32 | 59.89 | 67.70 | 66.60 | 67.07 | 50.51 | 38.10 | 54.53 | 49.39 | 49.12 | 41.58 | 47.21 |
| FedDISC | 74.32 | 73.47 | 71.25 | 66.79 | 75.28 | 70.06 | 71.86 | 56.82 | 51.43 | 59.45 | 56.17 | 52.32 | 45.64 | 53.64 |
| FGL | 74.62 | 79.43 | 71.26 | 68.65 | 76.37 | 74.31 | 74.11 | 57.25 | 49.35 | 61.81 | 58.42 | 54.29 | 47.62 | 54.79 |
| FedLMG_FT | 75.13 | 73.30 | 70.31 | 68.88 | 73.60 | 72.51 | 72.29 | 54.63 | 49.21 | 58.13 | 54.75 | 54.64 | 47.03 | 53.07 |
| FedLMG_SD | **77.34** | **79.94** | **75.01** | **71.87** | **76.69** | **74.92** | **75.96** | **61.49** | **51.47** | **65.28** | **60.03** | **59.57** | **51.14** | **58.16** |
| FedLMG_MD | 74.66 | 75.78 | 71.05 | 69.58 | 74.34 | 72.11 | 72.92 | 54.42 | 47.83 | 59.85 | 53.94 | 52.96 | 45.15 | 52.36 |

medium of knowledge distillation, utilizing all client classifiers $\mathcal{F}_{\boldsymbol{\theta}_k}$ as teachers to distill their knowledge into the aggregated model. The loss function is as follows:

$$
\mathcal{L}_{agg}(\hat{\mathbf{x}}_i^k, y_i^k) = \mathcal{L}_{CE}(\mathcal{F}_{\boldsymbol{\theta}_g}(\hat{\mathbf{x}}_i^k), y_i^k) \\
+ w_{MD} KL(\mathcal{F}_{\boldsymbol{\theta}_g}(\hat{\mathbf{x}}_i^k) \| \frac{\sum_{k'=1}^{K} \mathcal{F}_{\boldsymbol{\theta}_{k'}}(\hat{\mathbf{x}}_i^k)}{K})
$$

(12)

where $\mathcal{F}_{\boldsymbol{\theta}_g}$ and $\mathcal{F}_{\boldsymbol{\theta}_h}$ are the aggregated model and client classifiers, $w_{MD}$ is the weight of distillation loss. This strategy maximally leverages the knowledge from all client classifiers. However, in cases of substantial variations among clients or under the label distribution skew, the teachers from different clients may provide wrong guidance, impacting the performance of the aggregated model.

**Specific-teacher Distillation.** The third strategy is specific-teacher distillation. Given that $\hat{\mathbf{x}}_i^k$ with its client ID $k$, we can use the specific teacher model $\mathcal{F}_{\boldsymbol{\theta}_k}$ to achieve model aggregation. The loss function is as follows:

$$
\mathcal{L}_{agg}(\hat{\mathbf{x}}_i^k, y_i^k) = \mathcal{L}_{CE}(\mathcal{F}_{\boldsymbol{\theta}_g}(\hat{\mathbf{x}}_i^k), y_i^k) \\
+ w_{SD} KL(\mathcal{F}_{\boldsymbol{\theta}_g}(\hat{\mathbf{x}}_i^k) \| \mathcal{F}_{\boldsymbol{\theta}_k}(\hat{\mathbf{x}}_i^k))
$$

(13)

The meanings of each parameter are the same as mentioned earlier. When there are significant differences between clients or under the label distribution skew, this strategy ensures accurate guidance and the stable performance of the aggregated model.

## 4. Experiments

### 4.1. Experimental Settings

**Datasets.** We conduct experiments on three large-scale real-world image datasets: **OpenImage** (Kuznetsova et al., 2020), **DomainNet** (Peng et al., 2019) and **NICO++** (Zhang et al., 2023c). **DomainNet** comprises six domains: *clipart, infograph, painting, quickdraw, real*, and *sketch*. Each domain has 345 categories. Following the partition in (Yang et al., 2024a;b), according to the hierarchy of categories provided by OpenImage, **OpenImage** is partitioned into 20 supercategories, with each supercategory comprising 6 fine-grained subclasses, serving as the six data domains for each category. **NICO++** involves 60 categories, with each category having six common domains shared across categories and six unique domains specific to each category. These two scenarios are respectively referred to as the **Unique NICO++** and **Common NICO++** datasets. For instance, in Common NICO++, both the *Cat* and *Dog* categories encompass 6 data domains: *autumn, dim, grass, outdoor, rock*, and *water*. In Unique NICO++, the *Cat* class comprises 6 unique data domains: *Eating, In Cave, In Mud, Jumping, Maine Cat*, and *Walking*, while the *Dog* class comprises 6 distinct data domains: *Lying, Pug Dog, Running, Sticking Out Tongue, Teddy Dog*, and *Wearing Clothes*.

**Compared Methods.** We primarily compare 3 strategies of FedLMG: Fine-Tuning (FedLMG_FT), Multi-teacher Distillation (FedLMG_MD), and Specific-teacher Distillation (FedLMG_SD) against 3 kinds of methods: 1) **Ceiling.** The

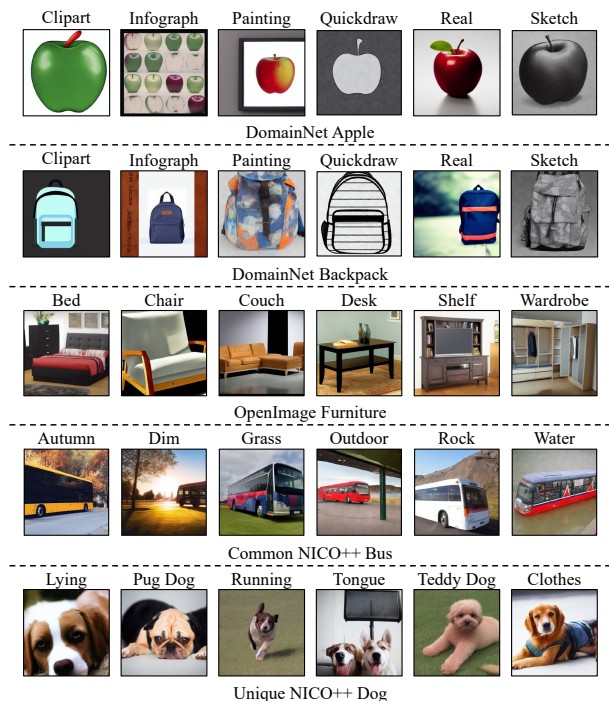

Clipart | Infograph | Painting | Quickdraw | Real | Sketch

DomainNet Apple

Clipart | Infograph | Painting | Quickdraw | Real | Sketch

DomainNet Backpack

Bed | Chair | Couch | Desk | Shelf | Wardrobe

OpenImage Furniture

Autumn | Dim | Grass | Outdoor | Rock | Water

Common NICO++ Bus

Lying | Pug Dog | Running | Tongue | Teddy Dog | Clothes

Unique NICO++ Dog

*Figure 2.* Visualization of generated samples on different datasets.

*Table 2.* Impact of different losses.

| BN Loss | CLS Loss | clipart | infograph | painting | quickdraw | real | sketch | average |
|---------|----------|---------|-----------|----------|-----------|------|--------|---------|
| | | 31.8 | 11.61 | 31.14 | 4.13 | 61.53 | 31.44 | 28.60 |
| ✓ | | 40.55 | 15.89 | 36.84 | 7.64 | 58.87 | 36.05 | 32.64 |
| | ✓ | 38.92 | 14.82 | 33.57 | 4.61 | 58.69 | 36.07 | 31.11 |
| ✓ | ✓ | **44.25** | **17.51** | **38.74** | **9.43** | 57.31 | **38.44** | **34.28** |

performance ceiling of traditional FL methods is centralized training, involving the uploading of all client local data for the training of the aggregated model. 2) Traditional FL methods with multiple rounds of communications: **FedAvg** (McMahan et al., 2017), **FedDF** (Lin et al., 2020), **FedProx** (Li et al., 2020b), **FedDyn** (Acar et al., 2021). All of them have 20 rounds of communications. Following standard experimental settings, each round involves one epoch of training on each client. And we use ImageNet as the additional public data for distillation in FedDF. 3) Diffusion-based OSFL methods: **FedDISC** (Yang et al., 2024a), **FGL** (Zhang et al., 2023a) and **Prompts Only**. Although FedDISC is designed for semi-supervised FL scenarios, we remove the pseudo-labeling process of FedDISC and directly utilize the true labels of client images. Another point to notice is the **Prompts Only**, where the server does not use the client models uploaded from clients at all but only uses the text prompts of category names in the server image generation. It is important to note that, we also compare our method with **DENSE** (Zhang et al., 2022) and **FedCVAE** (Heinbaugh et al., 2022). However, due to the reasons highlighted in the Introduction, these meth-

*Table 3.* Impact of the number of generated images.

| | clipart | infograph | painting | quickdraw | real | sketch | average |
|--|---------|-----------|----------|-----------|------|--------|---------|
| $N = 3450$ | 40.77 | 15.95 | 35.66 | 8.51 | 55.81 | 37.1 | 32.3 |
| $N = 10350$ | 44.25 | 17.51 | 38.74 | 9.43 | 57.31 | 38.44 | 34.28 |
| $N = 17250$ | **46.03** | **18.61** | **40.07** | **10.7** | **59.27** | **40.72** | **35.9** |

ods primarily demonstrate results on smaller datasets like CIFAR-10. Therefore, these methods are not utilized here.

### 4.2. Main Results

Firstly, we conduct experiments to assess our method under feature distribution skew. From the results in Table 1, we highlight several observations: 1) Compared to all used methods, FedLMG consistently demonstrates superior performance across all datasets, which effectively demonstrates the performance of our method on large-scale realistic datasets. 2) Compared to Ceiling, in multiple data domains, our method exhibits superior performance, which confirms that the vast knowledge of diffusion models can effectively assist in the training of the aggregated model, resulting in the performance surpassing the traditional performance ceiling of centralized training. 3) Compared to Prompts Only, FedLMG shows promising performance on most clients, which emphasizes the necessity of assistance from the client models. Images generated solely based on text prompts have a distribution that is too broad and cannot comply with the local image distribution of the clients. 4) Compared to other diffusion-based methods, FedLMG demonstrates a performance advantage on most clients. This indicates that our method can extract more precise information about the client distribution from the client models, guiding the DM to generate higher-quality synthetic datasets. 5) The comparison of three model aggregation strategies shows that FedLMG_SD achieves superior performance on most clients, further confirming the ability of our method to generate data that complies with different distributions, enabling the specific teachers to provide more accurate guidance. Additionally, on DomainNet, the reason FedLMG_MD performs better on more clients is that the distribution within the same data domain is more complex, and there is more overlap between domains, allowing teachers from other clients to contribute valuable information.

To further validate the generating ability of our method, we present visualization results in Figure 2, illustrating that FedLMG successfully generates images that possess accurate semantic information and exhibit competitive quality with the original client datasets.

### 4.3. Ablation Experiments

**Ablation experiments about the loss functions.** Table 2 and Figure 3 demonstrate the impact of different losses. As

*Table 4.* Comparison about the communication cost.

| Parameters requiring communication (M) | | | | | | | | | | | | | | |
|---|---|---|---|---|---|---|---|---|---|---|---|---|---|---|
| Ceiling | | | FedAvg | | | FedDISC | | | FGL | | | FedLMG | | |
| Upload | Download | Total | Upload | Download | Total | Upload | Download | Total | Upload | Download | Total | Upload | Download | Total |
| 270.95 | 0 | 270.95 | 233.8 | 222.11 | 455.91 | 4.23 | 427.62 | 431.85 | 0.345 | 469.73 | 470.08 | 11.69 | 0 | **11.69** |

*Table 5.* Comparison of the client computation cost.

| | FedAvg | FedDISC | FGL | FedLMG |
|---|---|---|---|---|
| FLOPS (G) | 72.8 | 334.73 | 227.34 | **3.64** |

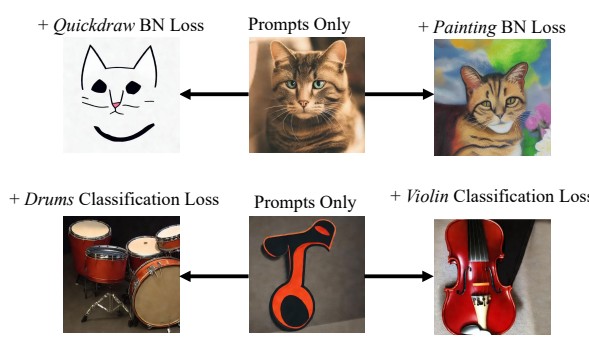

*Figure 3.* Visualization about the impact of different losses.

shown in Table 2, the inclusion of BN loss and classification loss introduces category and contextual information, leading to the generation of synthetic datasets that better comply with client distributions, thereby training an aggregated model with improved performance. Figure 3 illustrates that classification loss provides more precise category information, while BN loss introduces more detailed contextual information, preserving the stylistic alignment between the synthetic dataset and the client data.

**Ablation experiments about the number of generated images.** The number of images in the synthetic dataset directly affects the performance of the aggregated model. In most of our experiments, to ensure a fair comparison with the **Ceiling** baseline in terms of the number of training images, we maintain the total number of images across all clients equal to that in the synthetic dataset. However, to further investigate the impact of the number of generated images on the performance of the aggregated model, we conduct related ablation experiments. Using each client's local model, we generated 10, 30, and 50 images per class for the categories present on the client. Therefore, for DomainNet with 345 categories, the total number of images per client $N$ in the synthetic dataset is 3450, 10350, and 17250 respectively. The results are presented in Table 3. As shown, increasing the number of generated images leads to improved performance. Moreover, we observe that as the dataset size grows, the performance gain does not saturate, further demonstrating the diversity of the synthetic dataset.

**Ablation experiments about the number of clients** We conduct ablation experiments on the number of clients on the Common NICO++ and Unique NICO++ datasets under label distribution skew. We set the number of clients to 6, 30, 60, and 180. The experimental results are presented in Table 6. It can be observed that, with the well-trained client models, the quality of the synthetic dataset and the performance of the trained aggregated model are not significantly affected by the increase in the number of clients. What's more, our method is relatively independent for each client, showing strong adaptability to the increase in the number of clients. These results underscore the practicality of our method in scenarios with a large number of clients.

### 4.4. Discussions and Limitations

**Communication Cost.** We thoroughly discuss the communication cost of the proposed method. Since the communication cost of FedAvg, FedDF, FedProx, and FedDyn are essentially the same, their results are not repeated. Prompts Only does not involve any communication between the client and the server. The number of iterations and the used model structures follow the default experimental settings. The comparison results of the upload and download communication costs between FedLMG and other methods are shown in Table 4. From the results, it is evident that because there is no foundation model used on the clients, FedLMG does not involve any download communication cost, resulting in the lowest communication cost.

**Computation Cost.** The computation cost include the computation cost on the client and the server. Regarding the server computation cost, on one hand, as same as other diffusion-based OSFL methods, FedLMG involves generating synthetic datasets and training the aggregated model on the server, leading to similar server computation cost. Meanwhile, FedLMG uses local models for conditional generation rather than incorporating additional input conditions, which perform repeated noise prediction at each timestep. FedLMG can reduce the server computation cost required for generation. On the other hand, in FL, although generating data requires more computation on the server, the server, as the center of the FL, typically has sufficient computational power. Therefore, lower client computation cost are relatively more advantageous for practicality (Kairouz et al., 2021). Regarding client computation cost, we conduct thorough quantitative experiments in Table 5. Since FedAvg,

*Table 6.* Results of the ablation experiments on the number of clients.

| The Number of | Unique NICO++ | | | | | | | Common NICO++ | | | | | | |
|---|---|---|---|---|---|---|---|---|---|---|---|---|---|---|
| Clients | client0 | client1 | client2 | client3 | client4 | client5 | Avg | autumn | dim | grass | outdoor | rock | water | Avg |
| 6 | 75.13 | 73.30 | 70.31 | 68.88 | 73.60 | **72.51** | 72.28 | **54.63** | 49.21 | **58.13** | 54.75 | **54.64** | 47.03 | **53.07** |
| 30 | 74.24 | **74.32** | **71.63** | 68.97 | 72.53 | 72.10 | **72.29** | 53.23 | **50.47** | 57.07 | **55.66** | 54.09 | 46.49 | 52.84 |
| 60 | 74.52 | 73.64 | 70.39 | **69.54** | **73.65** | 71.05 | 72.13 | 54.27 | 49.18 | 57.99 | 54.44 | 53.16 | 48.90 | 52.99 |
| 180 | **75.64** | 74.06 | 70.89 | 68.67 | 72.15 | 71.94 | 72.23 | 53.26 | 49.45 | 56.55 | 53.86 | 53.19 | 46.73 | 52.17 |

Client Dataset | The Synthetic Dataset

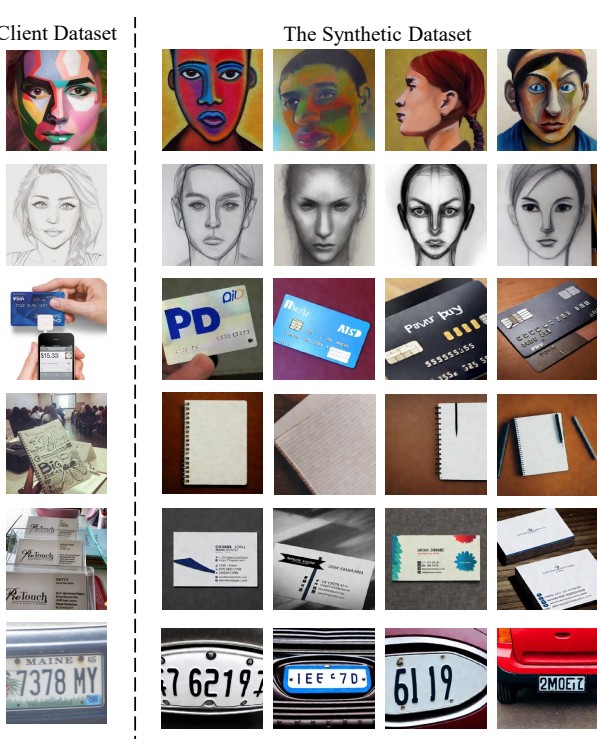

*Figure 4.* Visualization of privacy-sensitive categories.

FedDF, FedProx, and FedDyn have similar client computation costs, they are not compared separately. Ceiling and Prompts Only do not involve any client computation, so they are not included in the comparison. The number of iterations and the model structures used follow the default experimental settings. The quantified results demonstrate that FedLMG has a significant advantage in client computation cost due to not involving any foundation model on the clients and only requiring training of client models in a single round of iteration, demonstrating its practicality.

**Privacy Issues.** Transmitting client models is the most common practice in FL. Since our method only requires uploading the client model once, it offers a significant advantage in privacy protection compared to other traditional FL methods. Compared to other OSFL methods, where either the trained generative model (Zhang et al., 2022; Heinbaugh et al., 2022) or direct descriptions of client images are uploaded (Yang et al., 2024a; Zhang et al., 2023a), extracting

*Table 7.* Impact of adding noise to the uploaded parameters.

| Noise Weight | clipart | infograph | painting | quickdraw | real | sketch | average |
|---|---|---|---|---|---|---|---|
| 0 | **44.25** | **17.51** | **38.74** | **9.43** | **57.31** | **38.44** | **34.28** |
| 20 | 43.53 | 17.06 | 38.49 | 9.13 | 57.08 | 38.15 | 33.91 |
| 50 | 42.82 | 16.73 | 37.63 | 8.67 | 56.51 | 37.25 | 33.273 |
| 100 | 40.86 | 16.04 | 35.28 | 8.19 | 55.68 | 35.14 | 31.86 |

user privacy information from client models is more challenging. To further demonstrate FedLMG's performance in privacy protection, we conduct sufficient quantitation and visualization experiments. We select some categories from OpenImage that may contain privacy-sensitive information, such as human faces, vehicle registration plates, and notebooks. We train client models on these categories and generate synthetic datasets. The visualization results are shown in Figure 4. It can be observed that the synthetic datasets only share similar styles and accurate semantics with the original client datasets. It is almost impossible to extract specific privacy-sensitive information from the client models, which aim to learn the classification boundary.

Moreover, since our method only involves uploading local models, consistent with traditional FL, most existing privacy-preserving techniques in traditional FL can be directly applied to our method. To further validate this, we add noise to the uploaded model parameters to ensure differential privacy, which is commonly used in FL (Wei et al., 2020), and evaluate its impact on model performance. We add different weights of noises to the uploaded model parameters and conduct experiments. The results are presented in Table 7. As shown in this table, our method can effectively accommodate privacy-protecting techniques in FL. By adding noise to the uploaded parameters, we can protect client privacy and achieve differential privacy.

## 5. Conclusion

In this paper, we propose FedLMG. Compared to existing OSFL methods, we eliminate the need for auxiliary datasets and generator training, making it effortlessly applicable in real-world scenarios. Comprehensive experiments on three large-scale datasets demonstrate that the proposed FedLMG outperforms all compared methods and even surpasses the performance ceiling of centralized training in some cases, underscoring the potential of applying DMs in OSFL.

## Impact Statement

Given that our method utilizes a pre-trained diffusion model, there is a possibility of generating sensitive or private information. However, the Stable Diffusion model we rely on has been equipped with a robust safety-checking mechanism designed to minimize such risks. We also conduct sufficient experiments to demonstrate the performance of our method in privacy protection. As a result, we feel that our method does not raise additional significant concerns or potential broader impacts that warrant specific attention or further discussion here.

## Acknowledgements

This work was supported in part by the National Key R&D Program of China (No.2021ZD0112803), the National Natural Science Foundation of China (No.62176061), the Shanghai Research and Innovation Functional Program (No.17DZ2260900), and the Program for Professor of Special Appointment (Eastern Scholar) at Shanghai Institutions of Higher Learning.

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

In the appendix, we provide a substantial amount of content that couldn't be included in the main text due to space limitations, which can mainly be divided into three sections: 1) **Method Details.** We further elaborate on the details of the proposed method, including proofs of theoretical analyses and pseudo code of the proposed method. 2) **Experimental Setting Details.** More detailed information about the datasets and client partitions are provided, along with thorough implementation details. 3) **Supplementary Experiments.** We further demonstrate the performance of the proposed method through supplementary quantitation and visualization experiments, such as ablation experiments regarding the number of clients and the use of different diffusion models, quantitation and visualization experiments concerning privacy, and more visualization experiments of synthetic datasets, etc.

## A. Method Details

Our method focuses on one-shot federated learning (OSFL), involving using the client models trained by the clients to guide the DMs in generating the synthetic dataset that complies with different client distributions. In the main text, we delve into the relationship between the synthetic data distribution and the client distribution and provide theoretical analyses. In this section, we detail our theoretical analyses and provide the pseudocode of the proposed method in Algorithm 1.

### A.1. Proofs

In this section, we provide a proof of the theoretical analyses in the main text, regrading the upper bound of the KL divergence between the distributions of the synthetic dataset and the client datasets. Firstly, in section 3.1 of the main text, we introduce an assumption: for the data distribution of client $k$ $p_k(\mathbf{x})$ and the data distribution of the DMs $p_{\epsilon_\theta}(\mathbf{x})$, we have:

**Assumption 1** *There exists $\lambda > 0$ such that the Kullback-Leibler divergence from $p_k(\mathbf{x})$ to $p_{\epsilon_\theta}(\mathbf{x})$ is bounded above by $\lambda$:*

$$KL(p_{\epsilon_\theta}(\mathbf{x}) \| p_k(\mathbf{x})) < \lambda \tag{14}$$

Based on this assumption, considering the KL divergence between the distributions of the original client dataset $p_k(\mathbf{x})$ and the synthetic dataset $p_{\epsilon_\theta}(\mathbf{x}|\boldsymbol{\theta}_k)$, which is the conditional distribution of the DMs conditioned on the trained discriptions $\boldsymbol{\theta}_k$, we have:

**Theorem 1** *There exists $\lambda > 0$, for the local data distribution $p_k(\mathbf{x})$ and the conditional distribution $p_{\epsilon_\theta}(\mathbf{x}|\boldsymbol{\theta}_k)$ of the DM $\epsilon_\theta$ conditioned the client model $\mathcal{F}_{\boldsymbol{\theta}_k}$ trained on client $k$, we have:*

$$KL(p_k(\mathbf{x}) \| p_{\epsilon_\theta}(\mathbf{x}|\boldsymbol{\theta}_k)) < \lambda + \mathbb{E}(\log p_{\epsilon_\theta}(\boldsymbol{\theta}_k)) - \int p_k(\mathbf{x}) \log p_{\epsilon_\theta}(\boldsymbol{\theta}_k|\mathbf{x}) d\mathbf{x} \tag{15}$$

*Proof.* Firstly, based on the definition of KL divergence, we have:

$$KL(p_k(\mathbf{x}) \| p_{\epsilon_\theta}(\mathbf{x}|\boldsymbol{\theta}_k)) = -\int p_k(\mathbf{x}) \log \frac{p_{\epsilon_\theta}(\mathbf{x}|\boldsymbol{\theta}_k)}{p_k(\mathbf{x})} d\mathbf{x} \tag{16}$$

Based on the Bayes' theorem, we have:

$$p_{\epsilon_\theta}(\mathbf{x}|\boldsymbol{\theta}_k) = \frac{p_{\epsilon_\theta}(\boldsymbol{\theta}_k|\mathbf{x}) p_{\epsilon_\theta}(\mathbf{x})}{p_{\epsilon_\theta}(\boldsymbol{\theta}_k)} \tag{17}$$

From Eq. 16 and Eq. 17, we have:

$$
\begin{aligned}
KL(p_k(\mathbf{x}) \| p_{\epsilon_\theta}(\mathbf{x}|\boldsymbol{\theta}_k)) &= -\int p_k(\mathbf{x}) \log \frac{p_{\epsilon_\theta}(\mathbf{x}|\boldsymbol{\theta}_k)}{p_k(\mathbf{x})} d\mathbf{x} \\
&= -\int p_k(\mathbf{x}) \log \frac{p_{\epsilon_\theta}(\boldsymbol{\theta}_k|\mathbf{x}) p_{\epsilon_\theta}(\mathbf{x})}{p_k(\mathbf{x}) p_{\epsilon_\theta}(\boldsymbol{\theta}_k)} d\mathbf{x} \\
&= \int p_k(\mathbf{x}) \log \frac{p_k(\mathbf{x}) p_{\epsilon_\theta}(\boldsymbol{\theta}_k)}{p_{\epsilon_\theta}(\boldsymbol{\theta}_k|\mathbf{x}) p_{\epsilon_\theta}(\mathbf{x})} d\mathbf{x} \\
&= \int p_k(\mathbf{x}) \log \frac{p_k(\mathbf{x})}{p_{\epsilon_\theta}(\mathbf{x})} d\mathbf{x} + \int p_k(\mathbf{x}) \log \frac{p_{\epsilon_\theta}(\boldsymbol{\theta}_k)}{p_{\epsilon_\theta}(\boldsymbol{\theta}_k|\mathbf{x})} d\mathbf{x} \\
&= KL(p_k(\mathbf{x}) \| p_{\epsilon_\theta}(\mathbf{x})) + \int p_k(\mathbf{x}) \log \frac{p_{\epsilon_\theta}(\boldsymbol{\theta}_k)}{p_{\epsilon_\theta}(\boldsymbol{\theta}_k|\mathbf{x})} d\mathbf{x}
\end{aligned}
\tag{18}
$$

From Eq. 14 and Eq. 18, we have:

$$
KL(p_k(\mathbf{x}) \| p_{\epsilon_\theta}(\mathbf{x}|\boldsymbol{\theta}_k)) < \lambda + \int p_k(\mathbf{x}) \log \frac{p_{\epsilon_\theta}(\boldsymbol{\theta}_k)}{p_{\epsilon_\theta}(\boldsymbol{\theta}_k|\mathbf{x})} d\mathbf{x}
\tag{19}
$$

, where $\lambda$ is defined in Assumption 1. Next, we focus on the integral term within Eq. 19:

$$
\begin{aligned}
&\int p_k(\mathbf{x}) \log \frac{p_{\epsilon_\theta}(\boldsymbol{\theta}_k)}{p_{\epsilon_\theta}(\boldsymbol{\theta}_k|\mathbf{x})} d\mathbf{x} \\
&= \int p_k(\mathbf{x}) \log p_{\epsilon_\theta}(\boldsymbol{\theta}_k) d\mathbf{x} - \int p_k(\mathbf{x}) \log p_{\epsilon_\theta}(\boldsymbol{\theta}_k|\mathbf{x}) d\mathbf{x} \\
&= \mathbb{E}(\log p_{\epsilon_\theta}(\boldsymbol{\theta}_k)) - \int p_k(\mathbf{x}) \log p_{\epsilon_\theta}(\boldsymbol{\theta}_k|\mathbf{x}) d\mathbf{x}
\end{aligned}
\tag{20}
$$

Therefore, from Eq. 19 and Eq. 20, we have :

$$
KL(p_k(\mathbf{x}) \| p_{\epsilon_\theta}(\mathbf{x}|\boldsymbol{\theta}_k)) < \lambda + \mathbb{E}(\log p_{\epsilon_\theta}(\boldsymbol{\theta}_k)) - \int p_k(\mathbf{x}) \log p_{\epsilon_\theta}(\boldsymbol{\theta}_k|\mathbf{x}) d\mathbf{x}
\tag{21}
$$

With Theorem 1 proven, we thoroughly demonstrate that under the assistance of client models, the conditional distribution of the server's DM is sufficiently close to the local distribution of different clients. Moreover, the more comprehensive the client models training, the higher overlap between the non-conditional distribution of the diffusion model and the client local distributions, and the smaller the KL divergence between the conditional distribution and the local distributions. This provides a theoretical foundation for us to generate synthetic datasets that match different client distributions on the server.

## B. Experimental Setting Details

In this section, we detail the experimental settings of the proposed method that couldn't be elaborated on in the main text due to the space limitations, primarily comprising three parts: 1) **Dataset Details.** 2) **Client Partition Details.** 3) **Implementation Details.**

### B.1. Dataset Details

Our experiments are conducted on three datasets: **DomainNet** (Peng et al., 2019), **OpenImage** (Kuznetsova et al., 2020) and **NICO++** (Zhang et al., 2023c). The example images of each dataset are presented in figure 5. As mentioned in the main text, this figure clearly illustrates the emphases on the partition of data domains across different datasets is different. DomainNet primarily focuses on image style, OpenImage concentrates on fine-grained subcategories within each supercategory, Common NICO++ prioritizes image backgrounds, and Unique NICO++ places its emphasis on specific object attributes. The datasets we employ comprehensively simulate various types of differences that may exist among clients, thereby further enhancing the practicality of the proposed method.

---

**Algorithm 1** *FedLMG*: a heterogeneous one-shot **Fed**erated learning method with **L**ocal model-**G**uided diffusion models

---

**Input**: The client models $\{\mathcal{F}_{\boldsymbol{\theta}_k}\}_{k=1}^{K}$. A pre-trained diffusion model $\epsilon_\theta$. **Output**: An aggregated model $\mathcal{F}_{\boldsymbol{\theta}_g}$ adapted to the data distributions of all client datasets $\mathcal{D}^k = \{(\mathbf{x}_i, y_i)\}_{i=1}^{N_k}, y_i \in \{1, \ldots, \mathcal{C}\}$.

 1: create empty synthetic dataset $\hat{\mathbf{X}} = \{\}$
 2: **for** domain label $k = 1, \ldots, K$ **do**
 3:    **for** number of synthetic images $i = 1, \ldots, N$ **do**
 4:       randomly select a category $y$ supported by the client model $\mathcal{F}_{\boldsymbol{\theta}_k}$.
 5:       randomly sample the initial noise $\hat{\mathbf{x}}_T^M$ from $\mathcal{N}(0, \mathcal{I})$.
 6:       **for** $m = M, \ldots, 0$ **do**
 7:          use $\epsilon_\theta$ to compute $\epsilon_\theta\left(\hat{\mathbf{x}}_T^m, T | y\right)$
 8:          $\hat{\mathbf{x}}_{0,T}^m \leftarrow \frac{\hat{\mathbf{x}}_T^m - \sqrt{1-\alpha_T}\epsilon_\theta(\hat{\mathbf{x}}_T^m, T|y)}{\sqrt{\alpha_T}}$
 9:          $\mathcal{L}(\hat{\mathbf{x}}_T^m, y, \boldsymbol{\theta}_k) \leftarrow \mathcal{L}_{BN}(\hat{\mathbf{x}}_{0,T}^m, \boldsymbol{\theta}_k)$
10:          $\hat{\mathbf{x}}_T^{m-1} \leftarrow \hat{\mathbf{x}}_T^m - \eta \bigtriangledown_{\hat{\mathbf{x}}_T^m} \mathcal{L}(\hat{\mathbf{x}}_T^m, y, \boldsymbol{\theta}_k).$
11:       **end for**
12:       $\mathbf{x}_T \leftarrow \hat{\mathbf{x}}_T^0$
13:       **for** $t = T, \ldots, 0$ **do**
14:          use $\epsilon_\theta$ to compute $\epsilon_\theta\left(\mathbf{x}_t, t | y\right)$
15:          $\hat{\mathbf{x}}_{0,t} \leftarrow \frac{\mathbf{x}_t - \sqrt{1-\alpha_t}\epsilon_\theta(\mathbf{x}_t, t|y)}{\sqrt{\alpha_t}}$
16:          compute $\mathcal{L}(\mathbf{x}_t, y, \boldsymbol{\theta}_k)$ by Eq. (6).
17:          compute $\hat{\epsilon}\left(\mathbf{x}_t, t | y\right)$ by Eq. (8).
18:          compute $\mathbf{x}_{t-1}$ by Eq. (2).
19:       **end for**
20:       $(\hat{\mathbf{x}}_i^k, y_i^k) \leftarrow (\mathbf{x}_0, y)$
21:       add $(\hat{\mathbf{x}}_i^k, y_i^k)$ to synthetic dataset $\hat{\mathbf{X}}$
22:    **end for**
23: **end for**
24: use $\hat{\mathbf{X}} = \{(\hat{\mathbf{x}}_i^k, y_i^k)\}_{i=1}^{N}, k = 1, \ldots, K$ to train an aggregated model $\mathcal{F}_{\boldsymbol{\theta}_g}$ by fine-tuning, multi-teacher distillation or specific-teacher distillation as described in the main text.
25: **return** the aggregated model $\mathcal{F}_{\boldsymbol{\theta}_g}$

---

## B.2. Client Partition Details

### B.2.1. CLIENT PARTITION.

The client partitioning is primarily aimed at reflecting the non-IID of data across various clients. In federated learning, there are primarily two types of non-IID data: feature distribution skew and label distribution skew (Kairouz et al., 2021). We address these two scenarios separately in our client partition. For feature distribution skew, we conduct experiments on all four datasets. For each dataset, we allocate the six data domains of all categories to six clients, with each client possessing data from one unique domain for all categories. Regarding label distribution skew, experiments are conducted on Common NICO++ and Unique NICO++ datasets. Each 10 categories of the total 60 categories is grouped, resulting in six clients. Each client owns all data from 10 categories. As mentioned in the main text, there is no data overlap between clients in all partitions. Therefore, our partitioning maximizes the degree of non-IID among client data and considers various non-IID scenarios.

### B.2.2. NUMBER OF IMAGES.

The number of images on each client is important in our experimental setting, since we need to compare the performance of the proposed method with **Ceiling**, the performance ceiling of centralized training, involving the direct comparison between the synthetic dataset and original client dataset. Considering the cost of generation, we set the number of images generated with the assistance of each client model to 30 in most experiments except the ablation experiments about the number of images. The total number of images in each category of the synthetic dataset is 180. To ensure the fairness in comparing the synthetic dataset with the original client dataset, the maximum number of images for each category in each client local

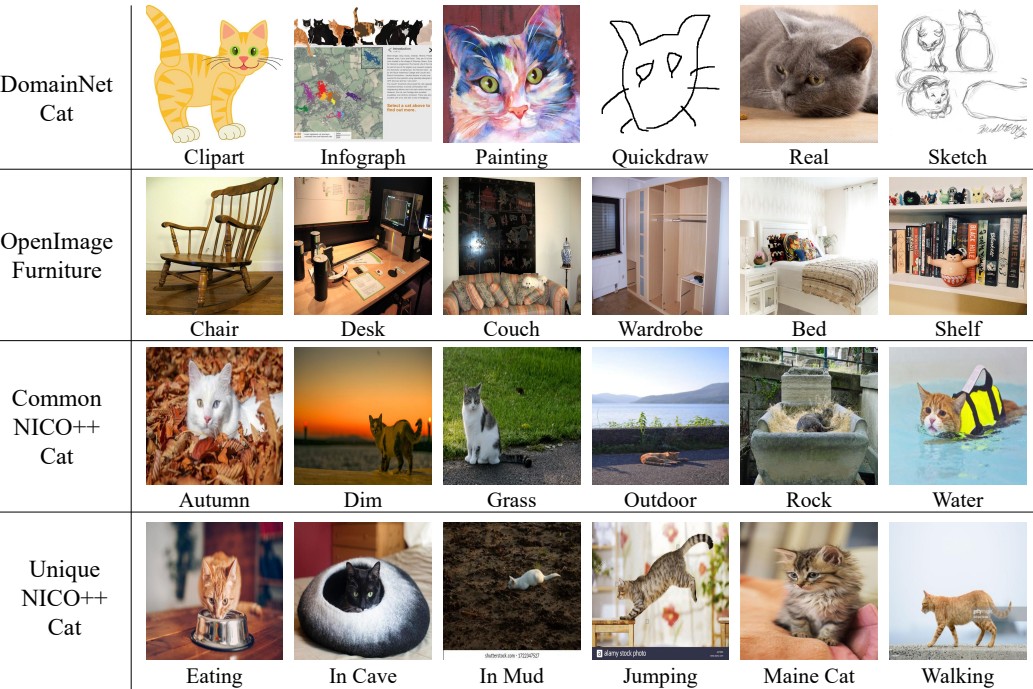

*Figure 5.* Example images of used datasets.

*Table 8.* Client partition on the OpenImage dataset.

| Supercategory | Baked Goods | Bird | Building | Carnivore | Clothing | Drink | Fruit | Furniture | Home appliance | Human body |
|---|---|---|---|---|---|---|---|---|---|---|
| Client0 | Pretzel | Woodpecker | Convenience Store | Bear | Shorts | Beer | Apple | Chair | Washing Machine | Human Eye |
| Client1 | Bagel | Parrot | House | Leopard | Dress | Cocktail | Lemon | Desk | Toaster | Skull |
| Client2 | Muffin | Magpie | Tower | Fox | Swimwear | Coffee | Banana | Couch | Oven | Human Mouth |
| Client3 | Cookie | Eagle | Office Building | Tiger | Brassiere | Juice | Strawberry | Wardrobe | Blender | Human Ear |
| Client4 | Bread | Falcon | Castle | Lion | Tiara | Tea | Peach | Bed | Gas Stove | Human Nose |
| Client5 | Croissant | Sparrow | Skyscraper | Otter | Shirt | Wine | Pineapple | Shelf | Mechanical Fan | Human Foot |
| Supercategory | Kitchen Utensil | Land Vehicle | Musical Instrument | Office Supplies | Plant | Reptile | Sports Equipment (Ball) | Toy | Vegetable | Weapon |
| Client0 | Spatula | Ambulance | Drum | Pen | Maple | Dinosaur | Football | Doll | Potato | Knife |
| Client1 | Spoon | Cart | Guitar | Poster | Willow | Lizard | Tennis Ball | Balloon | Carrot | Axe |
| Client2 | Fork | Bus | Harp | Calculator | Rose | Snake | Baseball | Dice | Broccoli | Sword |
| Client3 | Knife | Van | Piano | Whiteboard | Lily | Tortoise | Golf Ball | Flying Disc | Cabbage | Handgun |
| Client4 | Whisk | Truck | Violin | Box | Common Sunflower | Crocodile | Rugby Ball | Kite | Bell Pepper | Shotgun |
| Client5 | Cutting Board | Car | Accordion | Envelope | Houseplant | Sea Turtle | Volleyball | Teddy Bear | Pumpkin | Dagger |

dataset is also set to 30, as same as the image number for each category in each data domain of the synthetic dataset.

### B.2.3. NUMBER OF CLIENTS.

Regarding the number of clients, it's worth noticing that in our method, each client is entirely independent of the other clients. When the total number of images of the synthetic dataset is consistent, increasing the number of clients do not introduce interference between clients and affect the performance of the proposed method. Therefore, in the majority of our experiments, we set the number of clients to 6.

Nevertheless, we still conduct experiments related to the number of clients to demonstrate the practicality of the proposed method for a large number of clients. Following the commonly used Dirichlet distribution in partitioning non-IID clients (Hsu et al., 2019), in the feature distribution skew scenario, for the 6 domains of the Common NICO++ and Unique NICO++ datasets, we sample 5, 10, and 30 clients per domain from $Dirichlet(\alpha = 1.0)$ according to the categories. Consequently, the total number of clients changes from the original 6 to 30, 60, and 180. The clients simultaneously exhibit feature distribution skew and label distribution skew. The number of images in each category of the client local dataset remains 30. To ensure fairness in comparison across different numbers of clients, the total number of images in each category of the synthetic dataset remains 180, shifting the number of the generated images guided by each client model from 30 to 6, 3, and 1. For example, when the number of clients is 180, each model trained on each client guides the generation of 1 image in the

Table 9. Results of the ablation experiments on the used diffusion model.

| The Used | Unique NICO++ | | | | | | | Common NICO++ | | | | | | |
|---|---|---|---|---|---|---|---|---|---|---|---|---|---|---|
| Diffusion Model | client0 | client1 | client2 | client3 | client4 | client5 | Avg | autumn | dim | grass | outdoor | rock | water | Avg |
| SD-1.5 | 77.34 | 79.94 | 75.01 | 71.87 | **76.69** | **74.92** | 75.96 | **61.49** | 51.47 | 65.28 | **60.03** | 59.57 | 51.14 | 58.16 |
| SD-2.1 | **78.56** | **80.43** | **75.80** | **72.79** | 76.57 | 74.21 | **76.39** | 60.13 | **52.01** | **66.03** | 59.60 | **59.88** | **52.46** | **58.35** |
| LDM | 75.50 | 77.11 | 75.34 | 70.14 | 72.37 | 72.25 | 73.79 | 58.74 | 50.42 | 65.78 | 58.3 | 55.85 | 50.85 | 56.65 |

Table 10. Comparison of the FID between datasets.

| | FID between Different Datasets | | | | | | |
|---|---|---|---|---|---|---|---|
| | Clipart | Infograph | *Painting* | Quickdraw | Real_B | Sketch | Synthetic |
| Real_A | 284.39 | 228.44 | 221.01 | 297.37 | 131.48 | 195.74 | 147.94 |

synthetic dataset on the server.

### B.3. Implementation Details

In our experiments, we mainly use ResNet-18 (He et al., 2016) as the model structure of the aggregated model. In the experiment with heterogeneous models, the structures of the client models are MobileNetV3 (Howard et al., 2019), ResNet18 (He et al., 2016), ResNet34 (He et al., 2016), MobileNetV2 (Sandler et al., 2018), VGG16 (Simonyan & Zisserman, 2014), and ShuffleNet (Zhang et al., 2018), and the structure of the aggregated model is ResNet50 (He et al., 2016). The pre-trained DM we mainly used is *Stable-diffusion-v1.5* from the *HuggingFace* model repository, which includes a corresponding CLIP text encoder used in our method to extract text features $f_c$ for the name of each category $c$. We also use *Stable-diffusion-v2.1* from the *HuggingFace* model repository and the pre-trained *Latent Diffusion Model* (Rombach et al., 2022) from *Github*. The *Stable-diffusion-v1.5* and *Stable-diffusion-v2.1* are pre-trained on the LAION-5B dataset (Schuhmann et al., 2022) and the *Latent Diffusion Model* is pre-trained on the LAION-400M dataset (Schuhmann et al., 2021). Both datasets are large-scale image-text paired datasets, covering a wide range of image distributions encountered in daily life to satisfy Assumption 1. All experiments are conducted with four NVIDIA GeForce RTX 3090 GPUs. Regarding specific hyperparameters, the weight $\lambda$ in the loss function is set to 0.2. The relevant hyperparameters for the diffusion generation process are set to their default values. The number of inference steps is 50, and the guidance scale of the generation is 3.

## C. Supplementary Experiments

We primarily conduct supplementary experiments targeting three aspects that are not detailed in the main text because of the space limitation: 1) **Ablation Experiments.** Experiments regarding the used pre-trained diffusion models, the heterogeneous client models and the experiments under the label distribution skew. 2) **Privacy-related Supplementary Experiments.** Experiments regarding the discussions in the main text, including experiments on privacy. 3) **More Visualization Experiments.** Experiments to further illustrate the quality and diversity of the synthetic dataset.

### C.1. Ablation Experiments.

As stated in the main text, to further demonstrate the performance of the proposed method, we conduct sufficient ablation experiments. We discuss the two components of the loss function: the BN loss and the cross entropy loss, as well as the role of initial noise editing. Additionally, we discuss the impact of various hyperparameters used in our method, including the number of images generated by the server, the number of clients, and the specific employed DM. Due to space constraints in the main text, we provide some of the ablation experiments here.

#### C.1.1. EXPERIMENTS WITH DIFFERENT DIFFUSION MODEL

Since the synthetic datasets are used to train the aggregated model, the DM used for generating these synthetic datasets is important in our method. However, this does not mean that our method is heavily dependent on a specific DM. Firstly, as stated in Assumption 1, we only need a partial overlap between the distributions of the DM and the client distributions. This can be easily achieved with DM pre-trained on large-scale image datasets like LAION-5B (Schuhmann et al., 2022). Even if clients are concentrated in specialized fields, such as medical images, it is entirely feasible to firstly train the specialized DM on the server. Secondly, compared to other diffusion-based OSFL methods (Yang et al., 2024a; Zhang et al., 2023a) or

*Table 11.* Performance of different methods on NICO++ with heterogeneous models under feature distribution skew.

| | Unique NICO++ | | | | | | | Common NICO++ | | | | | | |
| --- | --- | --- | --- | --- | --- | --- | --- | --- | --- | --- | --- | --- | --- | --- |
| | client0 | client1 | client2 | client3 | client4 | client5 | Avg | autumn | dim | grass | outdoor | rock | water | Avg |
| *Ceiling* | *83.07* | *84.89* | *84.11* | *80.85* | *84.76* | *84.47* | *83.69* | *70.25* | *60.98* | *70.28* | *68.45* | *68.01* | *59.97* | *66.33* |
| FedDF | 71.09 | 70.57 | 67.96 | 66.67 | 62.11 | 52.34 | 65.12 | 49.97 | 43.37 | 60.05 | 55.12 | 53.75 | 51.17 | 52.24 |
| Prompts Only | 76.56 | 76.43 | 77.08 | 70.05 | 75.78 | 71.77 | 74.61 | 66.68 | 50.52 | 67.42 | 59.73 | 62.50 | 52.42 | 59.88 |
| FedDISC | 80.15 | 77.53 | 77.18 | 72.92 | 76.55 | 73.85 | 76.36 | 65.93 | 52.78 | 68.58 | 60.35 | 64.09 | 54.20 | 60.98 |
| FGL | 79.47 | 78.76 | 78.73 | 71.88 | 77.86 | 70.59 | 76.21 | 65.31 | 54.38 | 70.29 | 62.67 | 60.28 | 53.08 | 61.01 |
| FedLMG_FT | **81.77** | 76.56 | 79.68 | 75.03 | 76.82 | **77.24** | 77.85 | 65.06 | 57.33 | 69.47 | 63.64 | 66.01 | 57.34 | 63.14 |
| FedLMG_SD | 81.38 | 79.68 | 82.03 | 76.63 | 79.42 | 74.31 | 78.91 | 64.62 | 55.51 | 66.42 | 61.89 | 62.40 | 55.26 | 61.01 |
| FedLMG_MD | 81.64 | **81.91** | **82.55** | **81.89** | **82.94** | 75.48 | **81.07** | **68.19** | **57.55** | **71.46** | **66.28** | **67.18** | **58.82** | **64.91** |

*Table 12.* Performance of different methods on NICO++ under label distribution skew.

| | Unique NICO++ | | | | | | | Common NICO++ | | | | | | |
| --- | --- | --- | --- | --- | --- | --- | --- | --- | --- | --- | --- | --- | --- | --- |
| | client0 | client1 | client2 | client3 | client4 | client5 | Avg | autumn | dim | grass | outdoor | rock | water | Avg |
| *Ceiling* | *74.02* | *78.90* | *79.68* | *74.47* | *77.34* | *77.47* | *76.98* | *50.24* | *54.36* | *63.35* | *64.82* | *61.99* | *65.09* | *59.98* |
| FedAvg | 34.96 | 58.98 | 38.41 | 63.41 | 45.44 | 59.76 | 50.16 | 18.23 | 27.79 | 36.32 | 52.42 | 37.96 | 39.24 | 35.33 |
| FedDF | 51.85 | 52.34 | 55.85 | 52.47 | 54.42 | 59.24 | 54.36 | 31.40 | 32.22 | 43.73 | 45.19 | 36.01 | 43.08 | 38.61 |
| FedProx | 54.55 | 60.51 | 54.05 | 58.34 | 55.69 | 57.78 | 56.82 | 37.31 | 35.95 | 42.78 | 48.92 | 41.07 | 47.53 | 42.26 |
| FedDyn | 55.29 | 59.71 | 56.68 | 61.74 | 48.99 | 61.31 | 57.29 | 36.83 | 37.85 | 45.21 | 51.38 | 42.74 | 44.36 | 43.06 |
| Prompts Only | 67.38 | 71.88 | 67.70 | 64.19 | 63.41 | 63.28 | 66.31 | 38.64 | 45.55 | 53.08 | 54.72 | 50.19 | 59.91 | 50.35 |
| FedDISC | 71.89 | 73.20 | 70.51 | 70.02 | 75.62 | 69.82 | 71.84 | 50.75 | 51.64 | 60.79 | 58.33 | 55.41 | 57.28 | 55.70 |
| FGL | 69.51 | 74.59 | 71.36 | 69.41 | 69.65 | **71.42** | 70.99 | 45.34 | 51.41 | 60.44 | 59.65 | 58.87 | 62.33 | 56.34 |
| FedLMG_FT | **73.30** | 71.48 | 68.97 | 69.71 | 72.91 | 65.49 | 70.31 | **58.98** | 46.53 | **60.93** | 57.45 | 53.92 | 54.32 | 55.36 |
| FedLMG_SD | 67.77 | **76.04** | **73.95** | 70.44 | **76.56** | 68.35 | **72.19** | 49.27 | **52.63** | 57.52 | **59.84** | **65.11** | **64.98** | **58.25** |

federated learning methods based on foundation models (Su et al., 2024; Yang et al., 2023; Qiu et al., 2023), our method does not utilize the foundation models on the clients. Therefore, there is no requirement for the foundation model to fit the client data, significantly reducing the dependency on the foundation model.

To substantiate this claim, we conduct ablation experiments on three commonly used diffusion models, *Stable-diffusion-v1.5*, *Stable-diffusion-v2.1*, and *Latent Diffusion Model* (LDM). The experimental results are provided in Table B.2.1. From the table, it is evident that: 1) Our method can train high-performance aggregated models with different diffusion models. 2) Although the performance with LDM consistently surpasses other traditional FL methods, there is a performance gap compared to using Stable Diffusion. This is mainly because the LDM is pre-trained on LAION-400M (Schuhmann et al., 2021), and the difference in data scale results in a less extensive distribution and limited generative capability. This indirectly supports Theorem 1. 3) The best training results are achieved using Stable-diffusion-v2.1. Despite both Stable-diffusion-v2.1 and Stable-diffusion-v1.5 being pre-trained on LAION-5B (Schuhmann et al., 2022), Stable-diffusion-v2.1 has better generative quality due to its parameters and improved denoising capability. These results indicate that our method is not limited to the specific diffusion model, enhancing its practicality.

### C.1.2. EXPERIMENTS ON HETEROGENEOUS CLIENT MODELS

In addition to data heterogeneity, we also discuss the performance of FedLMG when there is heterogeneity in model structures across clients. Since FedAvg, FedProx, and FedDyn all require averaging the model parameters uploaded by clients and do not support heterogeneous model aggregation, they are not included in the comparison. The results are shown in the table B.3. From the table, we highlight the following two observations: 1) Because FedLMG does not restrict the specific model structure used by clients, these results also demonstrate excellent performance of our method on heterogeneous models, consistently surpassing all comparison methods and exceeding the performance ceiling of centralized training on some clients, further enhancing the practicality of FedLMG. 2) Notably, unlike the scenario with homogeneous client models, FedLMG_MD achieves better performance on most clients with heterogeneous models. The primary reason is that due to differences in model structures, the classifiers uploaded by clients vary significantly in performance. As a result, teachers from other clients might provide more accurate information despite the differences among clients, leading to better performances.

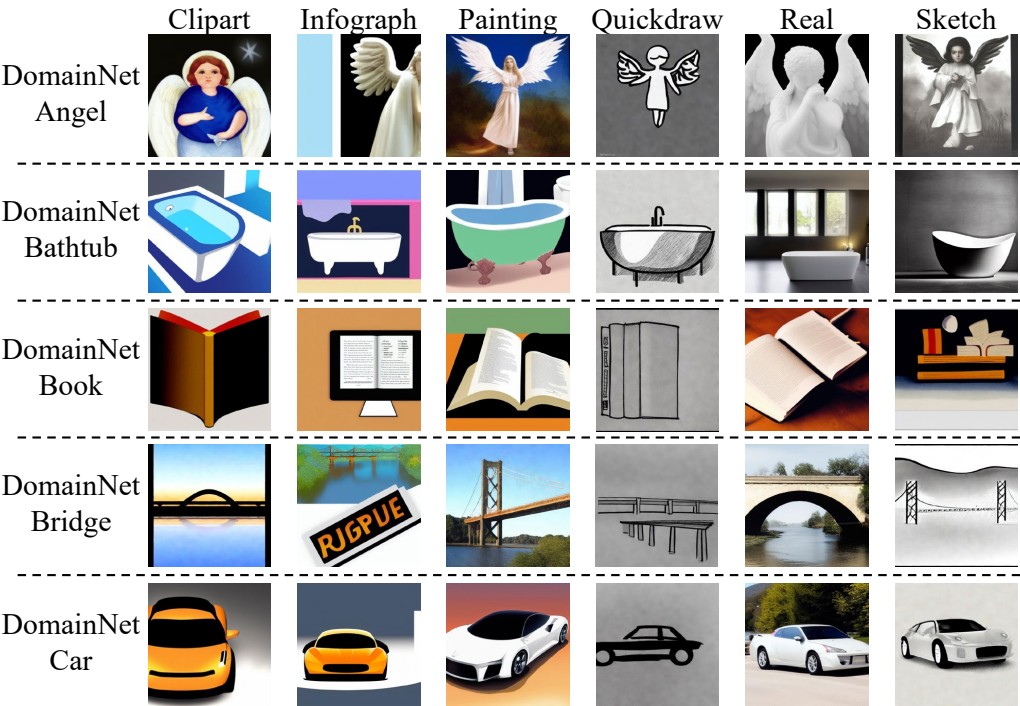

*Figure 6.* Supplementary visualization of the synthetic dataset on DomainNet.

### C.1.3. EXPERIMENTS UNDER LABEL DISTRIBUTION SKEW

To validate the performance of FedLMG under label distribution skew, we conduct experiments on Unique NICO++ and Common NICO++. Due to the label distribution skew among clients, the teachers from different clients cannot provide effective information, and therefore FedLMG_MD is not included in this comparison. The results are presented in Table B.3. As shown in the table, similar to the experiments conducted under feature distribution skew, our method also outperforms all comparison methods under label distribution skew, with performance on some clients even surpassing the centralized training upper bound. This further demonstrates the adaptability of our method on various non-IID client data scenarios.

### C.2. Privacy-related Supplementary Experiments.

In the main text, we conduct sufficient discussions and visualization experiments to demonstrate the privacy protection performance of our method. Here, we further address privacy issues, including comparing the FID (Fréchet Inception Distance) between the synthetic dataset and the client datasets, as well as presenting more visualization experiment results.

To illustrate the effectiveness of our method in preserving privacy, we assess the Fréchet Inception Distance (FID) between the synthetic dataset and the client local datasets, focusing on categories containing potentially sensitive information, such as the *Face* in DomainNet. Specifically, we divide the dataset of the *Real* domain into two non-overlapping parts, referred to as *Real_A* and *Real_B*, to represent the FID between datasets without privacy leakage but with the same style. We compute the FID between Real_A and datasets from other domains, as well as the Real domain of the synthetic dataset. Additionally, we gather photos of the same individual with different styles from the internet, partitioning them into two groups and calculating the FID between these groups to establish a threshold for potential privacy leakage. Through various experiments, we determine that this FID threshold is approximately 80, as shown in Table 10. Analysis of the table reveals that the FID between Real_A and the synthetic dataset falls within an optimal range—not too low, which could imply the presence of identical images leading to privacy concerns, nor too high, indicating significant style discrepancies in the generated images. These quantitative findings affirm that with the assistance of client models, the server can produce a synthetic dataset that aligns with client distributions while safeguarding privacy-sensitive information.

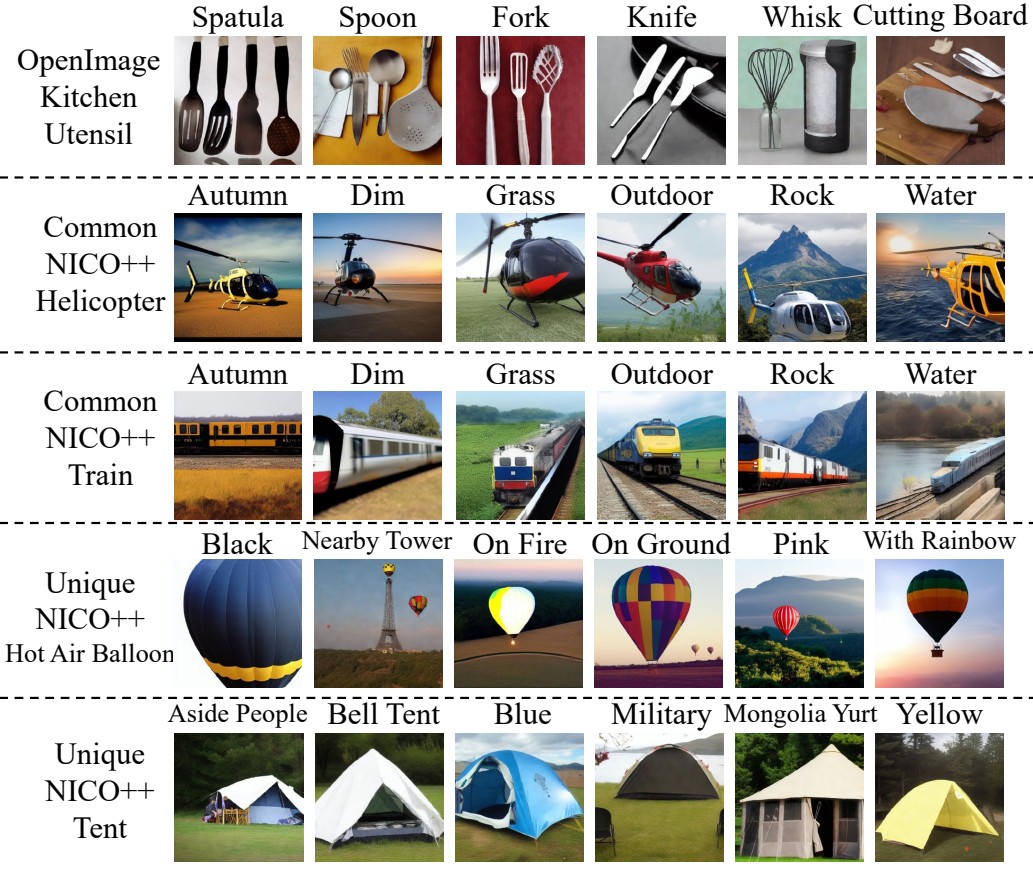

*Figure 7.* Supplementary visualization of the synthetic dataset on OpenImage and NICO++.

## C.3. More Visualization Experiments.

Similar to the main text, we conduct more visualization experiments to illustrate the quality and diversity of the synthetic dataset. The experimental results are presented in Figure 6 and 7. These visualizations further demonstrate that the generated synthetic dataset complies with various client distributions in style, subcategory, or background, with accurate semantic information. The synthetic dataset has comparable diversity and quality with the original client datasets, directly contributing to the performance of the trained aggregated model.

