# OpenReview forum: "One-Shot Heterogeneous Federated Learning with Local Model-Guided Diffusion Models"
_ICML.cc/2025/Conference — ICML 2025 poster_

### Official Review · Reviewer_xTsg · 2025-03-14

**Overall Recommendation:** 4

**Summary:**

This paper addresses the important and interesting problem of one-shot federated learning (OSFL), aiming to reduce the communication round of FL to 1. With the help of pretrained Classifier-Guided Diffusion Models, this paper proposes to generate local clients' data distribution in the server side with the guidance of the locally update models. The generated data are further used to train an aggregated global model.

## update after rebuttal

After two iteractions with the Authors, my questions has been solved.

I would recommend **Accept** after the necessary revisions, including more accurate explanations to the theretical results and proper citations for the borrowed contents. This paper addresses the interesting and important topic of one-shot federated learning by proposing a effective and reasonable method of utilizing classifier-guided DM. Even though it adds more computational requirements to the server comparing with standard federated learning method, the benefits from single communication round worth the cost. After all, if your computaional ability can not even run a diffusion model, you are not qualified as a server.

**Claims And Evidence:**

Clear and Convincing.

**Essential References Not Discussed:**

BN loss has been widely used, for example, [Yin'2020], while the related citations are missing.

[Yin'2020] Yin, Hongxu, et al. "Dreaming to distill: Data-free knowledge transfer via deepinversion." CVPR 2020.

**Experimental Designs Or Analyses:**

Experimental designs and analyses totally make sense.

**Methods And Evaluation Criteria:**

Make sense.

**Other Comments Or Suggestions:**

Typos.

- Theorem 1: error in right column of line 187.
- Eq 7, error in s (not bold)
- Eq 8, $\hat{\boldsymbol{s}}_{0,t}$

  or

  $\hat{\boldsymbol{s}}_{0}$?
- Right column of line 373, table 4.2?
- In line 437, it is claimed that the local model training only takes 1 iteration. I guess you mean a single round of optimization.

**Other Strengths And Weaknesses:**

Strength:

- This paper addresses the important and interesting setting, one-shot FL, which restricts communication round of FL to only 1.

- The proposed method generates synthetic images in the server side for aggregated model training, which makes sense considering computational ability.

- Even though this paper does not propose novel techniques, the creative combination of existing techniques to address the important and cutting-edge problem makes this paper interesting.

- Experiments are adequate and solid, especially the datasets are realistic and large-scale. Performance improvements are significant.

- I like the Privacy Issues part, which explains why the proposed method will not recover the original client data using meaningful experimental results, i.e. avoiding raising privacy concerns. It should also be highlighted in the abstract and introduction, avoiding confusion in the beginning.

Weaknesses:

- The explanation of theorem 1 is not enough. Refer to questions to authors below.

- BN loss only works with models with BN layers, while it does not work on others such as transformers or models for other data modalities.

**Questions For Authors:**

Theorem 1: the explanation of the last two terms are not clear enough.

1. Why is $\mathbb{E}(\log p_{\epsilon_\theta}(\boldsymbol{\theta}_k))$ a constant?

    What does $p_{\epsilon_\theta}(\boldsymbol{\theta}_k)$ represent?

2. Why is that minimizing the negative log-likelihood is equivalent to minimizing the cross-entropy loss?

**Relation To Broader Scientific Literature:**

The proposed method effectively addressed the one-shot federated learning problem, with experimental results showing significant performance improvements. The proposed method is intersting and totally makes sense, providing remarkable contribution to soloving the one-shot federated learning problem.

**Theoretical Claims:**

I have checked the proofs.
However, the proof for Theorem 1 is not enough, specifically the explanation of the last two two terms in equation 5. See the questions for authors below.

---

> ### Author Rebuttal · Authors · 2025-03-31
>
> We sincerely appreciate your recognition of our work and your valuable feedback. Below, we provide detailed responses to the key concerns you raised:
>
> >**(Essential References Not Discussed)** *"BN loss has been widely used, for example, [Yin'2020], while the related citations are missing."*
>
> Thank you for pointing this out. In the revised version of our manuscript, we will incorporate relevant references[1,2,3] to ensure a more comprehensive discussion of BN Loss and its applications.
>
> >**(Generality of BN Loss)** *Weaknesses #1 "BN loss only works with models with BN layers, while it does not work on others such as transformers or models for other data modalities."*
>
> We refer to the loss function in our study as BN Loss primarily because BN has been widely adopted, as demonstrated in prior works [1,2,3]. However, the core mechanism of BN Loss involves leveraging statistics, such as mean and variance, which are also present in other normalization layers such as Layer Normalization [4]. Notably, Layer Normalization has been extensively used in transformer, including models like CLIP [5]. Therefore, our method is not strictly limited to models with BN layers but can be extended to transformers and other architectures employing various normalization layers.
>
> >**(Further Explanation of Theorem 1)** *Weaknesses #1 "The explanation of theorem 1 is not enough."*
>
> >* *"Why is $\mathbb{E}(\log p\_{\epsilon\_\theta}(\boldsymbol{\theta}\_k))$ a constant?"*
>
> This term is solely dependent on the parameters of the diffusion model, which remain fixed during our method. Consequently, $\mathbb{E}(\log p_{\epsilon_\theta}(\boldsymbol{\theta}\_k))$ can be considered a constant.
>
> >* *"What does $p\_{\epsilon_\theta}(\boldsymbol{\theta}\_k)$ represent?"*
>
> This term represents the default data distribution of the diffusion model. After extensive pretraining, $p\_{\epsilon\_\theta}(\boldsymbol{\theta}\_k)$ approximates the data distribution of the diffusion model’s pre-train dataset.
>
> >* *"Why is that minimizing the negative log-likelihood is equivalent to minimizing the cross-entropy loss?"*
>
> The negative log-likelihood and the cross-entropy loss are formally equivalent. When the target distribution is a one-hot distribution, maximizing the likelihood corresponds to minimizing the cross-entropy loss. This equivalence is widely utilized in deep learning, particularly in the loss functions of softmax-based classification tasks[6].
>
> >**(Typos)** *Other Comments Or Suggestions*
>
> We appreciate your meticulous attention to detail in identifying typographical errors. These will be corrected in the revised manuscript. Additionally, we will conduct a thorough review to ensure the clarity and accuracy of our presentation.
>
> Once again, thank you for your insightful review and constructive feedback. We look forward to any further comments you may have.
>
> 	[1] Dreaming to distill: Data-free knowledge transfer via deepinversion, CVPR 2020.
> 	[2] Are Large-scale Soft Labels Necessary for Large-scale Dataset Distillation? NIPS 2024.
> 	[3] Source-Free Domain Adaptation for Semantic Segmentation, CVPR 2021.
> 	[4] Layer Normalization, NIPS 2016.
> 	[5] Learning transferable visual models from natural language supervision, ICML 2021.
> 	[6] Machine learning: a probabilistic perspective, MIT press 2012.

---

> > ### Comment · Reviewer_xTsg · 2025-04-05
> >
> > The questions to the theorem 1 remain unsolved after the rebuttal.
> > - The last two terms in Eq~ 5 do not make sense.
> > - The probability $\log p_{\epsilon_\theta}(\boldsymbol{\theta}_k)$ in the second term represent the distriubtion of model parameters in the diffusion model, while the parameters of pretrained diffusion model are deterministic not random. I also referred to the original paper proposing classifier guided diffusion model [Dhariwal & Nichol, 2021], where there are no such notations.
> > - The third term is totally different from cross-entropy loss, especially the third term measures the divergence between data distribution and parameters distribution, which does not make any sense.
> > - Most importantly, note that cross-entropy itself is not upper bounded, leaving the KL divergence in Eq~5 not upper-bounded.
> >
> > ---------------
> >
> > **After author's reply to the rebuttal comment above.**
> >
> > I apologize for the typo. Let me rephrase the questions as follow.
> >
> > - What does $p_{\epsilon_\theta}(\boldsymbol{\theta}_k)$ represent?
> >
> >    Since $\theta_k$ denotes the parameters of local trained model, it is irrelevant to the DM $\epsilon_{\theta}$.
> >
> > - Since the expectation in the second term is w.r.t. the local data distribution $p_k(\boldsymbol{x})$ and $\theta_k$ is trained based on the local data, is the second term still constant?
> >
> > - The explanation to the third term is somwhat convincing, i.e. even though this term is not computable, it possibly represents the mismatch between local data distribution and local model parameters.
> >
> > - Most importantly, this paper borrows content heavily from the paper **FedDEO**, especially the theoretical part, where the only difference is replacing the original $\boldsymbol{d}$ with $\boldsymbol{\theta}_k$. However, the related work FedDEO is not properly cited in the context of the borrowed content, which significantly degrades the quality of this paper.
> >
> > ------------------------
> > **After second iteraction with Authors**
> >
> > My questions has been solved.
> > - After dropping the $\epsilon_\theta$ from both marginal and conditional distribution of local model parameters $\theta_k$, the two terms make more sense than before.
> > - For the second term, I would recommend carefully explaining it as intuitively local data distribution absolutely makes impact on local model. One possible explanation would be that given the local data distribution, fixed model parameters initialization and optimizer, the locally optimized model parameters remain fixed after fixed number of iterations.
> > - I agree that this paper has significant difference from FedDEO. As mentioned before, proper citation is extremely important.
> >
> > I would recommend **Accept** after the revisions, including more accurate explanations to the two terms and proper citations. This paper addresses the interesting and importatnt topic of one-shot federated learning by proposing a effective and reasonable method of utilizing classifier-guided DM. Even though it adds more computational requirements to the server comparing with standard federated learning method, the benefits from single communication round worth the cost. After all, if your computaional ability can not even run a diffusion model, you are not qualified as a server.

---

> > > ### Author Response · Authors · 2025-04-06
> > >
> > > ---
> > > We sincerely apologize for the typo in our initial rebuttal, which may have caused a misunderstanding of our method. We appreciate your careful reading and now provide clarifications and detailed responses below:
> > >
> > > > * *"What does $p\_{\epsilon_\theta}(\boldsymbol{\theta}_k)$ represent?"*
> > >
> > > In our paper, $p\_{\epsilon_\theta}(x)$ denotes the default data distribution learned by the DM. $p\_{\epsilon\_\theta}(\boldsymbol{\theta}\_k)$ refers to the distribution of local model parameters, and $p\_{\epsilon\_\theta}(\boldsymbol{\theta}\_k|\mathbf{x})$ represents the conditional distribution of local model parameters given the client data $\mathbf{x}$.
> > >
> > > Since the latter two distributions are not related to the diffusion model parameters $\epsilon\_\theta$, it is more appropriate to express them as $p(\boldsymbol{\theta}\_k)$ and $p(\boldsymbol{\theta}\_k|\mathbf{x})$. We thank you for pointing out this inaccuracy, and we will revise the notation accordingly in the next version to improve precision and clarity.
> > >
> > > > * *"... is the second term still constant?"*
> > >
> > > Yes. As in our work and many other FL settings, once the local models are uploaded to the server, their parameters remain fixed during aggregation. Since the conditional distribution of the synthetic dataset $p_{\epsilon_\theta}(\mathbf{x}|\boldsymbol{\theta}_k)$ relies on the fixed local model parameters $\boldsymbol{\theta}_k$, the 2nd term is also constant in our analysis.
> > >
> > > > * *"... the related work FedDEO is not properly cited in the context of the borrowed content, which significantly degrades the quality of this paper."*
> > >
> > > We sincerely apologize for not explicitly citing FedDEO at the point of theoretical borrowing. During manuscript preparation, we indeed referred to some of FedDEO’s theoretical formulations to enhance the logical rigor of our paper. We will make the citation explicit and properly acknowledge their contribution in the revised version.
> > >
> > > It is important to emphasize that despite some theoretical similarities, our method differs significantly from FedDEO [1] in terms of practical design: **FedDEO requires training based on the DMs on the clients**, which obviously introduces substantial computation and communication costs. In contrast, our method significantly reducing the client burden. What's more, our method employs local models rather than additional prompts for guiding generation, eliminating the need for compositional diffusion, imposing a lower server computation cost. Below we provide a detailed comparison of model performance and computation costs between FedDEO, OSCAR[2] and our method, which will be included in the final version.
> > >
> > > ### Server computation cost :
> > >
> > > |          | FedDISC | FGL    | FedDEO | OSCAR  | FedLMG |
> > > |:--------:|:-------:|:------:|:------:|:------:|:------:|
> > > | flops (T)| 135.71  | 102.83 | 101.78 | 67.85  | **38.87** |
> > >
> > > ### Client accuracy comparison :
> > >
> > > |           | client0 | client1 | client2 | client3 | client4 | client5 | average |
> > > |:---------:|:-------:|:-------:|:-------:|:-------:|:-------:|:-------:|:-------:|
> > > | FedLMG_FT | 48.99   | 51.66   | 55.59   | 52.80   | 62.41   | 58.86   | 55.05   |
> > > | FedLMG_SD | 47.60   | **55.20** | **61.54** | 61.83   | 67.07   | **59.90** | **58.86** |
> > > | FedLMG_MD | 44.70   | 53.08   | 58.67   | 60.13   | 64.06   | 58.06   | 56.45   |
> > > | FedDEO    | **51.08** | 52.53   | 61.22   | **62.18** | 67.31   | 56.68   | 58.50   |
> > > | OSCAR     | 50.89   | 53.51   | 60.05   | 61.98   | **68.76** | 56.52   | 58.61   |
> > >
> > > ---
> > >
> > > We once again thank you for your thoughtful review and valuable feedback. Your comments helped us clarify critical aspects of our method and recognize areas where our explanations and citations can be improved. We will revise the corresponding parts accordingly, enrich the paper with further comparisons, and refine our writing to enhance the overall completeness and rigor. We sincerely hope that these improvements will better convey the contributions and practicality of our work.
> > >
> > > 	[1] FedDEO: Description-Enhanced One-Shot Federated Learning with DMs, MM 2024.
> > > 	[2] One-Shot Federated Learning with Classifier-Free Diffusion Models, ICME 2025.

---

### Official Review · Reviewer_5oFb · 2025-03-21

**Overall Recommendation:** 3

**Summary:**

This paper introduces FedLMG, a novel One-Shot Federated Learning (OSFL) method addressing limitations of diffusion model-based OSFL. FedLMG leverages locally trained client models to guide a server-side diffusion model in generating synthetic datasets tailored to individual client distributions. This approach eliminates the need for foundation models on clients, reducing computational burden and enhancing adaptability to heterogeneous clients. Extensive experiments on multiple datasets demonstrate FedLMG's superior performance over existing methods, even surpassing centralized training in some scenarios. Theoretical analysis and visualizations confirm the high quality and diversity of the generated synthetic data and the method's effectiveness in capturing client-specific distributions, highlighting the potential of diffusion models in practical OSFL.

**Claims And Evidence:**

The paper's central claim - the effectiveness and superiority of FedLMG for OSFL - is strongly supported by comprehensive evidence. Extensive quantitative experiments across diverse datasets (Table 1) convincingly demonstrate FedLMG's outperformance against various baselines, including traditional FL and other diffusion-based OSFL methods. The claim of surpassing centralized training ceilings is also empirically supported. Ablation studies (Table 4, Figure 3, Appendix C.1) provide evidence for the roles of BN loss and classification loss. Theoretical analysis (Theorem 1, Appendix A.1) offers a formal justification for the method's ability to generate client-aligned data. Visualizations (Figures 2, 4, 7, 8) qualitatively support the high quality and diversity of synthetic datasets and privacy-preserving nature.

**Essential References Not Discussed:**

No.

**Experimental Designs Or Analyses:**

The experimental designs are robust and effectively validate FedLMG. The core experiments (Table 1) comprehensively assess performance under feature distribution skew across diverse datasets, using appropriate baselines and metrics (accuracy). Ablation studies systematically dissect the contributions of key components like BN loss and diffusion model choices (Table 4, Appendix C.1), strengthening mechanistic understanding. The exploration of heterogeneous client models and label distribution skew (Appendix C.1) broadens the evaluation scope. Privacy experiments employing FID and visualizations (Figure 4, Appendix C.2) directly address privacy concerns. Visualization of synthetic data (Figures 2, 7, 8) provides qualitative validation of data quality and diversity.

**Methods And Evaluation Criteria:**

The paper proposes FedLMG, a novel method for one-shot federated learning utilizing diffusion models guided by locally trained client models. The methodology, encompassing local client training, guided synthetic data generation, and three aggregation strategies, is well-suited for addressing OSFL challenges, particularly in heterogeneous settings. The evaluation is comprehensive, employing large-scale datasets: OpenImage, DomainNet, and NICO++. Benchmarking against strong baselines, including traditional FL methods, diffusion-based OSFL methods, and a centralized training ceiling, provides a robust comparative analysis. Classification accuracy serves as a relevant and standard metric for evaluating model performance in image classification tasks within federated learning.

**Other Comments Or Suggestions:**

In this paper, a reference error occurred in Ablation Experiments in 4.3, that is, Table4.2 should be Table 4.

**Other Strengths And Weaknesses:**

Strengths:
1.FedLMG presents a new approach to OSFL by innovatively using locally trained client models to guide diffusion-based synthetic data generation.
2.The paper provides theoretical justification for the method, adding rigor and confidence to the empirical findings.
3. The method significantly enhances the practicality of OSFL by eliminating the need for foundation models on resource-constrained clients and effectively addressing heterogeneous client scenarios.

Weaknesses:
1.Although the paper provides a theory about KL divergence boundary, the specific selection of BN loss as a guiding mechanism lacks a strong theoretical basis.
2.Of the three polymerization strategies, FedLMG_SD (i.e. distillation using the synthetic sample and its corresponding client model) should theoretically give the best result, but in the experiment shown in Table 9 in the appendix, FedLMG_SD lags behind FedLMG_MD by a large margin. There is no relevant analysis in this paper.
3.Privacy assessments that rely on FID thresholds are not rigorous or convincing enough. The FID is not a specific privacy indicator, and the threshold chosen is subjective.

**Questions For Authors:**

1.Assumption 1's "boundedness" of KL divergence is overly broad and lacks quantitative validation, leading to a fragile theoretical foundation. If client data distributions differ significantly from the diffusion model's default distribution, FedLMG performance may substantially degrade.
2.The paper lacks quantitative analysis of the guidance signal effectiveness in image generation. The quality and effectiveness of client model guidance are unevaluated, obscuring FedLMG's working mechanism.
3.While the client side computation is reduced, server-side data generation can become a bottleneck for large-scale federated scenarios, limiting the actual application scale. The paper lacks a detailed breakdown of server-side computing costs and how they vary with dataset size.

**Relation To Broader Scientific Literature:**

FedLMG makes significant contributions to the intersection of Federated Learning (FL) and Diffusion Models (DMs). It addresses limitations of existing DM-based One-Shot FL (OSFL) methods (FedDISC, FGL) by eliminating the need for foundation models on clients. Unlike methods relying on public auxiliary data, FedLMG cleverly utilizes locally trained client models to guide DM-based data generation, a novel approach compared to generator-based OSFL and auxiliary information transfer methods. The distillation-based aggregation strategies build upon knowledge distillation in FL but introduce specific adaptations for OSFL with synthetic data.

**Theoretical Claims:**

The paper presents one main theoretical claim in Theorem 1, which is formally proven in Appendix A.1. I have carefully examined the provided proof of Theorem 1 and found it to be mathematically sound and logically consistent. The proof correctly demonstrates that, under Assumption 1 (bounded KL divergence between the diffusion model's unconditional distribution and client data distribution), the KL divergence between the synthetic dataset distribution and the client's local data distribution is indeed bounded.

---

> ### Author Rebuttal · Authors · 2025-03-31
>
> Thank you for recognizing our work. Below, we provide detailed responses to your concerns:
>
> >**(Server Cost)** *Questions #3"... server computing costs."*
>
> In FL, the server is generally designed to have sufficient resources to handle the model aggregation, but clients often exhibit significant heterogeneity, necessitating constraints on costs[1]. Our method adheres to this principle by reducing client burdens.
>
> Additionally, the following table presents a comparision of the server computation costs with other DM-based FL methods. Our method employs local models rather than additional prompts for guiding generation, eliminating the need for compositional diffusion, imposing a lower server computation cost.
>
> ||FedDISC| FGL|FedDEO|OSCAR|FedLMG|
> |:----:|:----:|:----:|:----:|:----:|:----:|
> |flops (T)|135.71|102.83|101.78|67.85|**38.87**|
>
> >**(Additional Ablation Experiments)** *Questions #4 "… vary with dataset size."*
>
> We appreciate the reviewer’s suggestion. The following table show that increasing the number of  generated images leads to an improvement in the performance. Moreover, we observe that the improvement does not saturate as the dataset size increases, further demonstrating the diversity of the synthetic dataset.
>
> |the number of generated images|clipart|infograph|painting|quickdraw|real|sketch|average|
> |:----:|:----:|:----:|:----:|:----:|:----:|:----:|:----:|
> |10|40.77|15.95|35.66|8.51|55.81|37.1|32.3|
> |30|44.25|17.51|38.74|9.43|57.31|38.44|34.28|
> |50|46.03|18.61|40.07|10.7|59.27|40.72|35.9|
>
> >**(Experimental Analysis)** *Weaknesses #2 "FedLMG_SD should give the best result ... no relevant analysis."*
>
> We would like to clarify that we include a dedicated analysis in line 937. We argue that due to the varying architectures of client models, some clients with more complex model structures demonstrate superior performance, allowing them to provide more accurate knowledge than the specific teacher in FedLMG_SD.
>
> >**(Privacy-Related Experiments)**: *Weaknesses #3 "… the threshold chosen is subjective."*
>
> We appreciate your suggestion, to further verify our method’s effectiveness in privacy protection, we employe [2] to ensure differential privacy and evaluate its impact on model performance. The results in the table below indicate that since our method only involves uploading local models, aligning with traditional FL, most privacy preserving methods in FL can be directly applied to our method without significantly degrading model effectiveness.
>
> |noise level $\epsilon$|clipart	|infograph|painting|quickdraw|real|sketch|average|
> |:----:|:----:|:----:|:----:|:----:|:----:|:----:|:----:|
> |0|**44.25**|**17.51**|**38.74**|**9.43**|**57.31**|**38.44**|**34.28**|
> |20|43.53|17.06|38.49|9.13|57.08|38.15|33.91|
> |50|42.82|16.73|37.63|8.67|56.51|37.25|33.273|
> |100|40.86|16.04|35.28|8.19|55.68|35.14|31.86|
>
>
> >**(Distribution Similarity)** *Questions #1 "Boundedness lacks quantitative validation ... client data distributions differ significantly..."*
>
> Quantitatively evaluating the similarity between the default distribution of a DM and client distributions is challenging, particularly given the large-scale pre-training datasets of DM. However, with the recent advancements, pre-trained DMs tailored for various domains[3,4] have become increasingly available. We believe that servers can select appropriate DMs based on the target application. Even where the data distribution is challenging, a pre-trained DM from a similar domain can be fine-tuned on the server. Thus, We assert that our method possesses practicality in diverse application scenarios.
>
> >**(BN Loss)** *Weaknesses #1 "... BN loss lacks a strong theoretical basis."*
>
> As noted by Reviewer #xTsg, BN Loss has been widely applied[5]. Since BN Loss compares statistics, it provides an intuitive guidance and is adopted in studies such as [5,6] without additional theoretical analysis. We plan to further explore its theoretical underpinnings in future to strengthen our method’s theoretical foundation.
>
> >**(Effectiveness of Guidance)** *Questions #2"... lacks quantitative analysis of the guidance effectiveness."*
>
> We would like to clarify that we provide quantitative analyses of the effectiveness of guidance. The Prompts Only in Table 1 represent results where no local model guidance was used, whereas FedLMG denotes results with guidance. We believe that the comparison between these settings sufficiently demonstrates the effectiveness of the guidance in our method.
>
> 	[1] A survey on federated learning: challenges and applications, IJMLC 2023.
> 	[2] Federated Learning with Differential Privacy: Algorithms and Performance Analysis, TIFS 2020.
> 	[3] Diffusion probabilistic models for 3d point cloud generation, CVPR 2021.
> 	[4] DiffuSeq: Sequence to Sequence Text Generation with DMs, ICLR 2023.
> 	[5] Dreaming to distill: Data-free knowledge transfer via deepinversion, CVPR 2020.
> 	[6] Are Large-scale Soft Labels Necessary for Large-scale Dataset Distillation? NIPS 2024.

---

### Official Review · Reviewer_A93d · 2025-03-22

**Overall Recommendation:** 3

**Summary:**

This paper introduces FedLMG, a novel approach for One-shot Federated Learning (OSFL) designed to establish an aggregated model within a single communication round. Specifically, FedLMG leverages fully-trained client models as classifier guidance to facilitate diffusion generation at the server. The generated images can represent the data distributions of the clients and are subsequently used for training an aggregated model. Experimental results demonstrate that FedLMG achieves superior performance compared to conventional Federated Learning (FL) and alternative diffusion-based OSFL methods, and sometimes even outperforms centralized training, which typically serves as the upper bound for FL.

**Claims And Evidence:**

In Line 99 (in the left column), the paper states “we propose FedLMG, a novel OSFL method, to
achieve real-world OSFL without utilizing any foundation models on the clients, ensuring no additional
communicational or computational burden compared to traditional FL methods.” However, FedLMG needs image generation with diffusion models on the server, which can be an additional computation cost because traditional Federated Learning does not require this step.

In Line 214 (in the left column), the paper mentions “Even if the clients specialize in certain professional domains, like medical images, it’s entirely viable to train specialized diffusion models on the server. Hence, this assumption is entirely reasonable, considering a comprehensive assessment of practical scenarios.”
However, within the context of Federated Learning, we do not know if the clients’ data are in specialized domains, and the server may not have the data and computation resources to train a specialized diffusion model. Therefore, rather than characterizing the aforementioned assumption as "entirely reasonable," it would be more accurate to consider it as a potential limitation of the proposed method.

**Essential References Not Discussed:**

Some related papers that follow the similar idea of using guidance for diffusion model generation in OSFL are missing, such as FedDEO [1], OSCAR [2], and FedCADO [3].

[1] 2024 ACM MM FedDEO: Description-Enhanced One-Shot Federated Learning with Diffusion Models

[2] 2025 arXiv One-Shot Federated Learning with Classifier-Free Diffusion Models

[3] 2023 arXiv One-Shot Federated Learning with Classifier-Guided Diffusion Models

**Experimental Designs Or Analyses:**

The experiments are conducted under standard Federated Learning settings, where heterogeneous class and style distributions are distributed across different clients. The performance metric is the accuracy of the aggregated model evaluated across all clients. In addition, the paper presents an ablation study on the classification and BN losses of the proposed method. Overall, the experimental design and analysis are considered valid.

**Methods And Evaluation Criteria:**

The proposed method, FedLMG, involves transmitting fully trained client models to a central server, where they are utilized to synthesize images for establishing an aggregated model. As the trained client model can be regarded as a compressed representation of its data distribution, the proposed method makes sense and is aligned with the setting of One-shot Federated Learning.

**Other Comments Or Suggestions:**

There are several typos from Line 178 and Line 188 (in the right column).
- Eq. 15 should be Eq. 5.
- Eq. 14 should be Eq. 4.
- It should be $p_k(x)$ instead of $p_n(x)$

In Appendix A, the proof is cut by Algorithm 1, making it a little hard to read.

**Other Strengths And Weaknesses:**

Strengths
- The paper is well-written and easy to follow. The proposed FedLMG also demonstrates promising performance in experimental evaluations.
- The paper addresses privacy concerns related to regenerating client data distributions using guided diffusion models and demonstrates, through visualization, that specific privacy-sensitive information may remain concealed and preserved.

Weaknesses
- The main idea of the paper lies in improving the use of diffusion models in OSFL by leveraging trained client classifiers as guidance and introducing the BN loss. However, the paper does not thoroughly discuss prior work on guided diffusion models [1][2], making it unclear whether the proposed BN loss is the most suitable choice for the OSFL setting or if existing methods could be equally applicable.
- A potential limitation arises from the requirement for some degree of overlap between the client data distributions and the diffusion model’s data distribution, as determined by the parameter $\lambda$ in Equation 4. Specifically, when clients have highly specialized data distributions, such as medical images, the diffusion model may struggle to reconstruct these distributions accurately due to a large $\lambda$ value. Although the paper suggests that a specialized diffusion model could be trained on the server to address this issue, doing so may not be feasible in practice due to limited data or computational resources.


[1] 2024 NeurIPS TFG: Unified Training-Free Guidance for Diffusion Models

[2] 2023 CVPR Universal Guidance for Diffusion Models

**Questions For Authors:**

Besides the weakness in the above sections, please also check the questions below:

1. Although the paper provides some visualizations suggesting that privacy-sensitive information may not be revealed by the guided diffusion generation, would applying noisy SGD (or related techniques) during client model training offer stronger privacy protection with formal differential privacy guarantees? How would this impact the performance of FedLMG?

2. How does the different degree of $\lambda$ impact the performance of FedLMG? For example, if clients hold data such as medical or aerial images, the diffusion model might struggle to reconstruct such distributions accurately. In this case, could FedAvg potentially perform better since it is indirectly trained on client data? Alternatively, how could FedLMG be adapted to handle such scenarios effectively?

3. Compared to FedAvg, which performs model aggregation through a simple averaging process, FedLMG requires significantly more computational resources on the server side due to image generation via diffusion models. What is the computational cost associated with this image-generation process? In Table 3, only the client-side cost is considered. How might the results change if the server-side cost is also taken into account?

**Relation To Broader Scientific Literature:**

This paper applies an existing idea of classifier-guidance diffusion models to the domain of One-shot Federated Learning (OSFL). The main contribution resides in the connection between these two distinct areas and the introduction of a BN loss to further improve the OSFL performance. Compared to existing OSFL methods such as FGL and FedDISC, a key advantage of the proposed FedLMG is that it eliminates the need for foundation model inference on clients, which may have limited computational resources.

**Theoretical Claims:**

In Theorem 1, the paper claims that the KL divergence between the client’s data distribution and the conditional distribution of the synthetic data is upper-bounded. Moreover, minimizing the cross-entropy loss within a client reduces the upper bound of the KL divergence. I’ve checked the proof in Appendix A, and the result seems to be correct.

---

> ### Author Rebuttal · Authors · 2025-03-31
>
> We appreciate your positive comments of our work and address each of your concerns as follows:
>
> >**(Server Cost)** *Claims #1 & Questions #3: "… DMs on the server, which can be an additional computation cost"*
>
> In FL, the server is generally designed to have sufficient resources to handle the aggregation of client models, but clients often exhibit significant heterogeneity, necessitating constraints on computation costs[1]. Our method adheres to this principle by reducing client burdens.
>
> Additionally, the following table presents a comparision of the server computation costs with other DM-based FL methods. Our method employs local models rather than additional prompts for guiding generation, eliminating the need for compositional diffusion, imposing a lower server computation cost.
>
> | |FedDISC| FGL|FedDEO|OSCAR|FedLMG|
> | :----: |:----: |:----: |:----: |:----: |:----: |
> |flops (T)	|135.71|102.83|101.78|67.85|**38.87**|
>
> >**(Uncertain Clients)** *Claims #2: "… we do not know if the clients’ data are in specialized domains"*
>
> The setting that the specific client tasks is uncertain for the server is Many-Task FL[2], where clients have diverse tasks simultaneously. However, this setting is not common. Most FL research presumes that the client task is known[1,3]. Therefore, we consider that our task setting is reasonable in real-world applications.
>
> >**(Applicability)** *Claims #2 & Weaknesses #2 & Questions #2: "… may not have the data and computation resources to train a specialized DM."*
>
> Currently, pre-trained DMs have been widely applied across various domains[4,5,6]. Based on the above discussion, we posit that the server know the overall task and can select an appropriate DM. Even where the data distribution is challenging, a pre-trained DM from a similar domain can be fine-tuned on the server. We assert that our method possesses practicality in diverse application scenarios.
>
> >**(References)** *Essential References: "... similar idea of using guidance for DM generation in OSFL are missing …"*
>
> We appreciate your suggestion. We incorporate the relevant works into our compared methods. As shown in the table below, our method achieves comparable performance without the employment of any foundational models on the clients, further demonstrating the performance of our method.
>
> ||client0|	client1|	client2|	client3|	client4|	client5|	average|
> | :----: |:----: |:----: |:----: |:----: |:----: |:----: |:----: |
> |FedLMG_FT|	48.99|	51.66|	55.59|	52.80|	62.41|	58.86|	55.05|
> |FedLMG_SD|	47.60|	**55.20**|	**61.54**|	61.83|	67.07|	**59.90**|	**58.86**|
> |FedLMG_MD|	44.70|	53.08|	58.67|	60.13|	64.06|	58.06|	56.45|
> |FedDEO	|	**51.08**|	52.53|	61.22|	**62.18**|	67.31|	56.68|	58.50|
> |OSCAR	|	50.89|	53.51|	60.05|	61.98|	**68.76**|	56.52|	58.61|
>
> >**(BN Loss)** *Weaknesses #1: "… whether the proposed BN loss is the most suitable choice"*
>
> As noted by Reviewer #xTsg, BN Loss has been widely utilized[7]. Although there is no prior precedent for employing BN Loss in the DMs, the general method of designing task-specific loss functions and guiding the diffusion process is well established[8]. While we acknowledge that BN Loss might not be the optimal choice in all circumstances, its effectiveness has been clearly validated by the ablation experiments presented in Table 4.
>
> >**(Privacy)** *Questions #1: "… applying noisy SGD during client model training offer stronger privacy protection"*
>
> We appreciate your suggestion. Since our method only involves uploading local models, aligning with traditional FL, most privacy preserving methods in FL can be directly applied to our method. To validate this, we adopt [8] to ensure differential privacy and evaluate its impact on model performance. The experimental results shown in the table below indicate that traditional FL privacy protection measures remain effective within our framework without significantly degrading model performance.
>
> |noise level $\epsilon$ |	clipart	|infograph|	painting|	quickdraw|	real|	sketch|	average|
> | :----: |:----: |:----: |:----: |:----: |:----: |:----: |:----: |
> |0|	**44.25**|	**17.51**|	**38.74**	|**9.43**	|**57.31**|	**38.44**	|**34.28**|
> |20|	43.53|	17.06|	38.49|	9.13|	57.08	|38.15	|33.91|
> |50|42.82|	16.73|	37.63	|8.67|	56.51	|37.25	|33.273|
> |100|	40.86	|16.04|	35.28|	8.19	|55.68	|35.14|	31.86|
>
> 	[1] A survey on federated learning: challenges and applications, IJMLC 2023.
> 	[2] Many-Task Federated Learning: A New Problem Setting and A Simple Baseline, CVPR 2023.
> 	[3] A survey on federated learning systems: Vision, hype and reality for data privacy and protection, TKDE 2021.
> 	[4] DMs in medical imaging: A comprehensive survey, MIA 2023.
> 	[5] DiffuSeq: Sequence to Sequence Text Generation with DMs, ICLR 2023.
> 	[6] DiffWave: A Versatile DM for Audio Synthesis, ICLR 2021.
> 	[7] Dreaming to distill: Data-free knowledge transfer via deepinversion, CVPR 2020.
> 	[8] Federated Learning with Differential Privacy: Algorithms and Performance Analysis, TIFS 2020.

---

### Official Review · Reviewer_vCBA · 2025-03-23

**Overall Recommendation:** 2

**Summary:**

In response to the increasing demand for efficient One-Shot Federated Learning (OSFL) solutions, this paper introduces FedLMG, a novel OSFL method leveraging Local Model-Guided diffusion models. Unlike existing OSFL methods that rely on foundation models deployed on client devices—causing significant computational overhead—FedLMG allows clients to train and upload only their local models, maintaining the lightweight nature of traditional Federated Learning (FL).

**Claims And Evidence:**

1. The privacy issue of the proposed approach remain questioned. Although the paper visualized the synthetic data, the limited discussions do not convince that the proposed method preserves user privacy. The proposed method highly relies on the generated synthetic dataset to present server-side model aggregation, which is naturally a negative part of diffusion-based FL that contradicts the privacy nature of FL.
2. The paper also claims that using the proposed method is computationally efficient. However, neither training the stable diffusion model to generate synthetic data nor the multi-teacher distillation process for knowledge aggregation is efficient. Compared with traditional FL, this raises much more burdens to the server. Also, does the proposed aggregation method via distillation introduces instability due to wrong teacher selection?

**Essential References Not Discussed:**

None.

**Experimental Designs Or Analyses:**

1. More ablation experiments are needed to prove the effectiveness of the proposed method, such as hyper-parameter studies, the size of synthetic data, the teacher selection constraints during distillation, etc.
2. The impact of the heterogeneity of datasets in the paper should be mentioned.
3. In Table 3, comparing the client computation costs between FedAvg and FedLMG to show that FedLMG is even more efficient than the traditional FedAvg does not make sense without simultaneously providing the convergence speed over communication rounds.

**Methods And Evaluation Criteria:**

1. The evaluations and discussions in the paper are only based on the image dataset. It is questioning how the proposed method will perform on non-image and other modality datasets, especially the privacy issues of generated data in the other modalities.
2. The computation costs comparison in the paper includes the communication cost and client computation costs. However, the cost of model aggregation on the server is not mentioned, which should be less-important but still necessary metric of the algorithm.

**Other Comments Or Suggestions:**

Typo "Table 4.2" -> Table 4 in section 4.3.

**Other Strengths And Weaknesses:**

1. Privacy discussions of the synthetic dataset are based on selected visualization results and FID scores. However, FID is not an official metrics for privacy protecting performance. How does it perform in terms of general quantitative metrics for privacy of FL, such as Gradient Leakage or Differential privacy, if applicable?

**Questions For Authors:**

The aggregated information from clients on server is represented as a synthetic dataset instead of an aggregated model like traditional FL.

**Relation To Broader Scientific Literature:**

The key contribution of the paper is in the knowledge aggregation part -- leveraging a synthetic dataset as the aggregated knowledge of all clients instead of leveraging a unified model as the aggregated knowledge. In terms of federated learning, this might be new.

However, the data synthesis associated with multi-teacher distillation is not new in domain generalization and knowledge distillation fields. The paper itself does not seem contribute significantly to the broader ML community.

**Theoretical Claims:**

No issues on proofs.

---

> ### Author Rebuttal · Authors · 2025-03-31
>
> We sincerely appreciate your review and valuable comments and provide detailed responses to the key concerns:
>
> >**(Privacy Concerns)** *Claims And Evidence #1 & Weaknesses "general quantitative metrics for privacy of FL, such as Gradient Leakage (GL) or Differential privacy (DP)."*
>
> Regarding GL, as discussed in [1] and [2], such attacks occur during the sharing of gradients , where attackers infer client data by analyzing gradients. However, our method does not involve sharing gradients, mitigating the risk of GL.
>
> Regarding DP, because our method only involves uploading local models, aligning with traditional FL, most DP preserving methods in FL can be directly applied to our method. To further validate this, we incorporated the method proposed in [3] to ensure DP and assessed its impact on performance. The following table demonstrate that the DP preserving methods remain applicable to our method without significantly compromising its effectiveness.
>
> |noise level $\epsilon$ |	clipart	|infograph|	painting|	quickdraw|	real|	sketch|	average|
> | :----: |:----: |:----: |:----: |:----: |:----: |:----: |:----: |
> |0|	**44.25**|	**17.51**|	**38.74**	|**9.43**	|**57.31**|	**38.44**	|**34.28**|
> |20|	43.53|	17.06|	38.49|	9.13|	57.08	|38.15	|33.91|
> |50|42.82|	16.73|	37.63	|8.67|	56.51	|37.25	|33.27|
> |100|	40.86	|16.04|	35.28|	8.19	|55.68	|35.14|	31.86|
>
> >**(Server Computation Cost)** *Claims And Evidence #2 & Methods And Evaluation Criteria #2: "... much more burdens to the server."*
>
> In FL, the server is generally designed to have sufficient resources to handle the aggregation of client models, but clients often exhibit significant heterogeneity, necessitating constraints on computation costs[5]. Our method adheres to this principle by reducing client burdens.
>
> Additionally, the following table presents a comparision of the server computation costs with other DM-based FL methods. Our method employs local models rather than additional prompts for guiding generation, eliminating the need for compositional diffusion, imposing a lower server computation cost.
>
> | |FedDISC| FGL	|FedDEO|	OSCAR|	FedLMG|
> | :----: |:----: |:----: |:----: |:----: |:----: |
> |flops (T)	|135.71	|102.83|	101.78|	67.85	|**38.87**|
>
> >**(Additional Ablation Experiments)** *Experimental Designs Or Analyses #1: "More ablation experiments are needed to prove the effectiveness of the proposed method."*
>
> We appreciate the reviewer’s suggestion. The following table show that increasing the number of  generated images leads to an improvement in the performance. Moreover, we observe that the improvement does not saturate as the dataset size increases, further demonstrating the diversity of the synthetic dataset.
>
> |the number of generated images| 	clipart	|infograph	|painting|	quickdraw|	real|	sketch	|average|
> | :----: |:----: |:----: |:----: |:----: |:----: |:----: |:----: |
> |10|	40.77|	15.95	|35.66|	8.51|	55.81|	37.1|	32.3|
> |30	|44.25	|17.51	|38.74	|9.43|	57.31	|38.44|	34.28|
> |50	|**46.03**|	**18.61**	|**40.07**|	**10.7**	|**59.27**	|**40.72**	|**35.9**|
>
> >**(Multimodality)** *Methods And Evaluation Criteria #1: "... how the proposed method will perform on non-image and other modality datasets."*
>
> Similar to many FL studies[5], we select image as the primary modality in our paper. However, our method is not restricted to the image modality. By utilizing DMs for other modalities such as [6, 7], our method can be seamlessly adapted to other modalities.
>
> >**(Dataset Heterogeneity)** *Experimental Designs Or Analyses #2: "The impact of the heterogeneity of datasets in the paper should be mentioned."*
>
> Dataset heterogeneity in FL primarily manifests as feature distribution skew and label distribution skew [8]. In Tables 1 and 10, we demonstrate the impact of both types of heterogeneities. Therefore, we respectfully disagree with this concern.
>
> >**(Contribution of the Paper)** *Relation To Broader Scientific Literature: "The key contribution of the paper is in the knowledge aggregation part..."*
>
> As stated in the Introduction and acknowledged by Reviewer #A93d and #5oFb, a key advantage of our method is eliminating the need for foundation model inference on clients. We believe this characteristic significantly enhances the practicality of diffusion-based FL methods and represents a meaningful contribution to the field.
>
>
> 	[1] Deep leakage from gradients, NIPS 2019.
> 	[2] Understanding Deep Gradient Leakage via Inversion Influence Functions, NIPS 2023
> 	[3] Federated Learning with Differential Privacy: Algorithms and Performance Analysis, TIFS 2020.
> 	[4] FedDEO: Description-Enhanced One-Shot Federated Learning with Diffusion Models, MM 2024.
> 	[5] A survey on federated learning: challenges and applications, IJMLC 2023.
> 	[6] DiffuSeq: Sequence to Sequence Text Generation with Diffusion Models, ICLR 2023.
> 	[7] DiffWave: A Versatile Diffusion Model for Audio Synthesis, ICLR 2021.
> 	[8] Federated Learning on Non-IID Data Silos: An Experimental Study, ICDE 2022.

---

### Decision · Program_Chairs · 2025-05-01

**Decision:**

Accept (poster)

**Comment:**

This paper studies the problem of One-shot Federated Learning by leveraging Local Model-Guided diffusion models.

Although the underlying techniques are not new, most of the reviewers think the proposed method is a good combination of existing techniques and the performance is promising.

Given all the reviews, I recommend weak accept of the paper and I encourage the authors to include the discussions and experiments in the response.